# Mechanistic Role of Jak3 in Obesity-Associated Cognitive Impairments

**DOI:** 10.3390/nu14183715

**Published:** 2022-09-09

**Authors:** Premranjan Kumar, Jayshree Mishra, Narendra Kumar

**Affiliations:** Department of Pharmaceutical Sciences, ILR College of Pharmacy Texas A&M Health Science Center, Kingsville, TX 78363, USA

**Keywords:** high-fat diet, obesity, Janus kinase-3, TREM-2, microglia, cognitive impairments

## Abstract

Background and Aims: A compromise in intestinal mucosal functions is associated with several chronic inflammatory diseases. Previously, we reported that obese humans have a reduced expression of intestinal Janus kinase-3 (Jak3), a non-receptor tyrosine kinase, and a deficiency of Jak3 in mice led to predisposition to obesity-associated metabolic syndrome. Since meta-analyses show cognitive impairment as co-morbidity of obesity, the present study demonstrates the mechanistic role of Jak3 in obesity associated cognitive impairment. Our data show that high-fat diet (HFD) suppresses Jak3 expression both in intestinal mucosa and in the brain of wild-type mice. Methodology: Recapitulating these conditions using global (Jak3-KO) and intestinal epithelial cell-specific conditional (IEC-Jak3-KO) mice and using cognitive testing, western analysis, flow cytometry, immunofluorescence microscopy and 16s rRNA sequencing, we demonstrate that HFD-induced Jak3 deficiency is responsible for cognitive impairments in mice, and these are, in part, specifically due to intestinal epithelial deficiency of Jak3. Results: We reveal that Jak3 deficiency leads to gut dysbiosis, compromised TREM-2-functions-mediated activation of microglial cells, increased TLR-4 expression and HIF1-α-mediated inflammation in the brain. Together, these lead to compromised microglial-functions-mediated increased deposition of β-amyloid (Aβ) and hyperphosphorylated Tau (pTau), which are responsible for cognitive impairments. Collectively, these data illustrate how the drivers of obesity promote cognitive impairment and demonstrate the underlying mechanism where HFD-mediated impact on IEC-Jak3 deficiency is responsible for Jak3 deficiency in the brain, reduced microglial TREM2 expression, microglial activation and compromised clearance of Aβ and pTau as the mechanism during obesity-associated cognitive impairments. Conclusion: Thus, we not only demonstrate the mechanism of obesity-associated cognitive impairments but also characterize the tissue-specific role of Jak3 in such conditions through mucosal tolerance, gut–brain axis and regulation of microglial functions.

## 1. Introduction

Cognitive impairment is a major characteristic of Alzheimer’s disease (AD), the major form of dementia. The incidences of AD are rising in the US, and its rate is expected to quadruple worldwide by 2050 [1,2,3]. Although between 2000 and 2013 death due to other diseases, such as stroke, heart diseases and prostate cancer, decreased by various percentages, death due to AD increased by 71% [4]. The deposition of β-amyloid (Aβ)-led neuroinflammation is a key factor in the pathogenesis and progression of AD [5], where both inflammation [6,7] and obesity [8,9] are documented characteristics in the AD population. Although the mucosal physiology of the intestine plays essential roles in both inflammation and obesity, little is known about its role in the cognitive impairment precursor of AD pathology. As both Aβ deposition and Tau hyperphosphorylation play key roles in brain inflammation and as drivers of AD [10], the impact of intestinal kinases on central-inflammation-led AD pathology nevertheless remains elusive.

Microglia are the resident innate immune cells of CNS and account for 10–15% of all cells found within the brain. Physically restricted to the brain upon completion of the blood–brain barrier, microglia become long-living, autonomous cell population that retains the ability to divide and self-renew throughout life without significant contribution from the circulating blood cells [11]. Microglial activation alters AD progression, where human genetics studies point to a causal role of microglial dysfunction in disease initiation [12]. Genome-wide association studies indicate several genes that increase the risk factors for AD [13,14,15] and are also associated with the regulation of microglial clearance of misfolded proteins, including Aβ and pTau [16,17].

Recent findings by others [18,19] and by our lab [20,21] support a key role for host genetic and dietary factors in chronic low-grade inflammation (CLGI) [22] in the gut as an early step in the onset of obesity. Previously, we showed that one such genetic factor is Janus Kinase 3 (Jak3), a non-receptor tyrosine kinase expressed in both immune cells and in intestine epithelial cells (IEC) of humans and mice [23,24,25]. Jak3 mediates signals initiated by the cell surface receptors [23,26], and Jak3-deactivating mutations lead to severe combined immunodeficiency [27,28]. In IEC [25], the essential roles of Jak3 include cytoskeletal remodeling [29], intestinal restitution [30], mucosal homeostasis [31], trans-molecular regulation of adapter protein Shc [32], and a deficiency of its functions leads to predisposition to chronic intestinal inflammation [20] and systemic CLGI-associated obesity [21]. As the modulation of Jak3 functions is implicated in different therapies [33,34], the drugs used to modulate Jak3′s immune functions are associated with severe infection and metabolic dysregulation [35,36,37,38,39,40,41], and Jak3 inhibition in patients with chronic inflammation [42,43,44] also leads to metabolic dysregulation. Previously, we reported that Jak3 regulated intestinal inflammation and predisposition to obesity-associated metabolic syndrome through the regulation of TLR-mediated mucosal tolerance [45]. However, the mechanisms through which Jak3 regulates the gut–brain axis and the associated cognitive functions during obesity are not known. In this report, we delineate the intestinal and brain functions of Jak3 and demonstrate how Jak3 deficiency not only affects obesity-associated gut dysbiosis but also the cognitive functions through microglial regulation of TREM2-medited activation and accumulation of Abeta and pTau in the brain.

## 2. Results

High-fat diet reduces the intestinal expression of Jak3, and an in vivo reconstitution of these conditions in mice leads to colonic dysbiosis and obesity-associated AD pathology. Previously, we reported that colonic mucosal Jak3 is decreased in human obese condition [46], and a deficiency of Jak3 expression in mice leads to exaggerated symptoms of obesity [21]. To determine the mechanism, as a first step, we determined whether HFD had an impact of Jak3 expression. Figure 1A (left panels) shows that HFD-fed mice not only had decreased expression of intestinal Jak3, but there was also a corresponding increase in the expression of intestinal Toll-like receptors (TLRs), TLR-2 and TLR-4. Since our previous study showed that Jak3 suppresses TLR expression through AKT pathways, we determined whether this was also true during HFD-mediated Jak3 suppression. Figure 1A (right panels) shows that HFD-mediated reduced Jak3 expression was associated with a corresponding increase in phosphorylated AKT (pAKT) and phosphorylated NFκB (pNFκB). For these experiments AKT, NFκB and β-actin were taken as control, which did not show changes as a result of HFD. Since HFD-mediated decreased Jak3 expression led to increased TLR expression in the intestine, we reconstituted these conditions using Jak3-KO mice and determined the impact of the deficiency of Jak3 expression on the changes in the gut microbiome composition, as determined through 16sRNA pyrosequencing of fecal DNA samples from co-housed WT and Jak3-KO siblings. The time course of change in the fecal microbiota shows that regardless of the familial connection, the microbial compositions were relatively similar in all mice at weaning (Figure 1B, top left and right). A deficiency of Jak3, however, was associated with a large subsequent shift in the relative abundance of specific taxa in fecal microbiota in a time-dependent manner, where Jak3-KO mice exhibited a statistically significant reduction in Bacteroidetes (Figure 1B, top left) and more than a 2-fold increase in Firmicutes (Figure 1B, top right). Among the Firmicutes, in particular, there was a 3-fold increase in the class Clostridia, order Clostridiales, in Jak3-KO mice. Among these, Jak3-KO mice had an approx. 3-fold increase in Lachnospiraceae (Figure 1B, bottom left) and over 2-fold increase in Ruminococcaceae families. Moreover, the time course showed that, although in the WT mice, there was a gradual decrease in the Firmicutes to Bacteroidetes ratio (F/B ratio), the Jak3-KO mice had a consistently increased F/B ratio (Figure 1B, bottom right) compared to WT mice. A metagenome-wide association study in humans [47] and intestinal colonization studies in mice [48] indicated that the Lachnospiraceae family contributes to the development of obesity and the associated co-morbidities. Since the deficiency of Jak3 affected the relative abundance of the major gut bacterial divisions derived from the common maternal inoculum, leading to gut dysbiosis, and the gut microbiome is reported to have a link with cognitive impairment [49], including neurological decline [10,50], we determined whether the deficiency of Jak3 impacted the cognitive functions in these mice. Figure 1C shows four parameters of cognitive assessments, viz., working memory errors (WME) (top left), roaming memory errors (RME) (top right), number of errors before a correct choice (EBCC) (bottom left) and reward memory errors (RWME) (bottom right). These parameters were measured using automated elevated radial arm maze equipped with MazesoftTM Software to calculate these parameters. A repeated-measure ANOVA on the performances indicated that age- and sex-matched wild-type (WT) littermate controls of Jak3-KO mice showed an average decrease in WME, EBCC and RWME over the sessions, suggesting that the animals had retained the spatial cognition task. A deficiency of Jak3 (Jak3-KO), however, resulted in a significant increase in the corresponding parameters, suggesting that the deficiency of Jak3 results in cognitive impairment, particularly with respect to the spatial cognition task. Moreover, there was a non-significant increase in RME in Jak3-KO mice compared to WT littermates.

Since accumulations of insoluble deposits of proteins β-Amyloid (Aβ) and hyper-phosphorylated (p)-Tau are the characteristic pathology features of cognitive decline in dementia patients, we determined whether deficiency of Jak3 expression led cognitive impairments were associated with Aβ and pTau accumulation under a normal diet (ND) and whether a high-fat diet (HFD) influenced such accumulation. Figure 1D (first and second column from the left) shows that WT littermate of Jak3-KO mice fed with ND expressed Jak3 in the brain; however, a deficiency of Jak3 expression led to the accumulation of Aβ and pTau in Jak3-KO mice. Since HFD suppressed Jak3 expression in the intestine, we determined whether HFD also had an impact on Jak3 expression in the brain. Figure 1D (third and fourth column from the left) shows that HFD-fed WT mouse also showed decreased Jak3 expressions in the brain, which were associated with increased Aβ and pTau accumulations (third column). Moreover, these accumulations were further accelerated in Jak3-KO mice (fourth column). The quantification of these data showed that the consumption of HFD led to a 3-fold decrease in Jak3 expression in the brain of WT mice, which was associated with a corresponding 15-fold increase in Aβ and a 20-fold increase in pTau. Moreover, the consumption of HFD led to accelerated accumulation of Aβ and pTau, where Jak3-KO mice showed an over 2-fold increase in both Aβ and pTau accumulation compared to ND-fed Jak3-KO mice. These results were further confirmed by Western blot (Figure 1E), which showed similar results.

Intestinal epithelial cell (IEC)-specific deficiency of Jak3 is responsible for the exaggerated symptoms of HFD-induced obesity and associated dysregulation in glycemic homeostasis. As a first step to reconstitute the obese human conditions of the intestinal deficiency of Jak3 expression, we generated floxed jak3 (jak3f/f), bred with Vil-cre mice, and determined whether Jak3 expressions were impacted in the intestine. Figure 2A shows that the expression of Jak3 was unaffected in the IEC of jak3f/f mice (upper panel; red), where Jak3 also colocalized with the IEC marker villin (green and merged upper panel: yellow). However, VilCre/jak3f/f mice showed a significant deficiency of Jak3 expression (lower panels) in the IECs, as indicated both by a deficiency of Jak3 expression and lack of co-localization of Jak3 with the IEC marker villin (merged lower panel showing only green). Since, apart from the IEC, Jak3 is also expressed in the immune cells, we wanted to reconfirm the IEC-specific deficiency of Jak3 expression. Western analysis using colon and spleen tissues lysates from jak3f/f and VilCre/jak3f/f mice showed that, although the splenic expression of Jak3 was unaffected in these mice (Figure 2B), Jak3 expression was nevertheless significantly lost in the colon of VilCre/jak3f/f mice when compared with jak3f/f controls, indicating an IEC-specific deficiency of Jak3 expression. For these experiments, western analysis using β-actin was taken as the control (bottom panel). Previously, we reported that a global deficiency of Jak3 (Jak3-KO) in mice leads to predisposition to obesity-associated metabolic syndrome [21]. Since our recent study indicated a significant deficiency of Jak3 in the intestine of obese humans [46], we determined whether the obesity-associated metabolic syndrome seen in Jak3-KO mice was due to an IEC-specific deficiency of Jak3. Figure 2C shows that the group of VilCre/jak3f/f mice with specific deficiency of Jak3 in IEC, when fed with a normal diet (Int-KO-ND), gained weight over time, which was comparable to the weight gained by the group of jak3f/f (control-2) mice, but under HFD, and the weights gained by both these groups were significantly higher than the group of jak3f/f (control-1) mice under a normal diet (FL-ND). Moreover, when the group of VilCre/jak3f/f mice with specific deficiency of Jak3 in IEC were fed with HFD, they gained 4.2-fold weight in the same period compared to control-2. To determine whether the IEC-specific deficiency of Jak3-led predisposition to obesity was associated with the dysregulation in glycemic homeostasis, we subjected the mice to a glucose tolerance test (GTT) through injecting them with a bolus of glucose and determined the restoration of blood glucose over a period of two hours. Figure 2D shows that although the group of FL-ND mice was able to restore the blood glucose level when subjected to a bolus of glucose, both FL-HFD and Int-KO-ND mice nevertheless had elevated blood glucose levels to start with compared to FL-ND mice and had failed to restore the blood glucose levels when subjected to GTT. Moreover, Int-KO HFD mice failed to restore blood glucose levels following GTT and had the highest blood glucose levels both before and after the GTT compared to all the other groups. These results show that IEC-specific deficiency of Jak3 not only led to a predisposition to HFD-led obesity but also to a compromise in their ability to restore glycemic homeostasis. Since obese humans had decreased intestinal Jak3, and HFD caused a deficiency of Jak3 expression in WT mice, we determined whether these effects were due to IEC-specific Jak3. The representative flow cytometry analysis and quantitation using the dot plots of individual mice in Figure 2E show that there was an over two-fold decrease in villin-Jak3 double positive cells in HFD-fed jak3f/f mice (E-1) compared to ND-fed jak3f/f (E-3) mice. Moreover, while ND-fed VilCre/jak3f/f mice (E-2) had a three-fold decrease in villin-Jak3 double positive cells (compared to E-1), feeding with HFD led to a four-fold decrease in villin-Jak3 double positive cells (E-4) in VilCre/jak3f/f mice compared to ND-fed jak3f/f mice. An average of the flow cytometry analyses from five independent experiments reconfirmed these results, as shown in Figure 2F.

IEC-specific deficiency of Jak3 is responsible for cognitive impairment and exaggerated symptoms of cerebral cortex accumulation of Aβ and pTau during HFD-induced obesity. Both HFD and obesity are associated with cognitive impairment throughout adulthood and increased dementia risk with aging, although the mechanisms are not fully understood [51]. HFD is a known cause of human obesity, and our previous report suggested decreased colonic mucosal expression of Jak3 during human obesity [46]. Since our data in Figure 1 showed HFD also decreased Jak3 expression in the intestine of WT mice, and a reconstitution of these conditions using global Jak3-KO mice led to cognitive impairment, we determined whether the intestinal tissue-specific IEC-Jak3 had an impact on Jak3 expression in the brain of mice fed with HFD. Representative brain immunofluorescence images in Figure 3A1 and the corresponding quantifications of the fluorescence intensities on the right panel show that although the expression of Jak3 in the brain of flox-Jak3 and IEC-Jak3-KO mice was unaffected upon feeding with ND, feeding with HFD caused a significant decrease in Jak3 expression, particularly in the IEC-Jak3-KO mouse brain. Western analysis using brain lysates in Figure 3A2 and the corresponding densitometric analysis (right panel) of the bands from these groups of mice further confirmed the immunofluorescence data, indicating the suppressive effects of HFD on Jak3 expression in the brain. Since our data showed that the mouse brain not only expressed Jak3 but that a global deficiency of Jak3 led to a predisposition to obesity and cognitive impairment, as a next step, we determined whether cognitive impairments, as seen in Jak3-KO mice, were related to the deficiency of IEC-specific Jak3. As with the global Jak3-KO, we assessed the four parameters of cognitive impairment in IEC-Jak3-KO mice, viz., WME, RME, EBCC and RWME, using automated elevated radial arm maze in intestinal epithelial-specific Jak3-KO mice and their littermate controls. Figure 3B1–4 show that the age- and sex-matched jak3f/f littermate controls (Flox-Control) of intestinal epithelial cell-specific Jak3-KO (IEC-Jak3-KO) mice showed a significant decrease in WME (B1), EBCC (B2), RWME (B3) and RME (B4) over the sessions, suggesting that the animals had retained the spatial cognition task. IEC-specific deficiency of Jak3 (IEC-Jak3-KO) nevertheless resulted in a significant increase in all the four corresponding parameters, viz., WME, RME, EBCC and RWME, of cognitive impairments over the testing sessions, suggesting that IEC-specific deficiency of Jak3 was responsible for cognitive impairment, as seen in global Jak3-KO mice, predominantly with respect to spatial cognition tasks. Next, we determined whether the intestinal Jak3 had an impact on the brain deposition of Aβ and pTau characteristics of dementia and whether HFD influenced such depositions. The brain immunofluorescence images for Aβ and pTau and the corresponding quantifications of the fluorescence intensities on the right panel in Figure 3C show that IEC deficiency of Jak3 expresion (IEC-Jak3-KO) resulted in increased accumulations of Aβ and pTau in the brains of mice fed with ND (second column from the left) compared to their littermate flox controls (first column). However, feeding with HFD led to a significant Aβ and pTau accumulation in the IEC-Jak3-KO groups of mice (fourth column).

The triggering receptors on microglial cells 2 (TREM2) interact with Jak3 in the brain, and IEC deficiency of Jak3 leads to an impairment of Jak3 interactions with TREM2 in the brain during HFD-induced obesity. TREM2 is a member of the TREM family of cell surface receptors protein exclusively expressed on the cell surface of microglia [52,53] and participating in diverse cell processes, including neurological development and inflammation. Since IEC deficiency of Jak3 impacted the cognitive impairment and brain deposition of pTau and Abeta, we determined whether IEC-Jak3 also impacted the TREM2 functions in the brain. To accomplish this, we first determined whether Jak3 interacted with microglial TREM2 in the brain. Figure 4A shows that TREM2 not only co-immunoprecipitated Jak3 in WT mouse brain (top panel, first lane), but Jak3 also co-immunoprecipitated TREM2 (middle panel, first lane) from the same brain lysates. As expected, the global deficiency of Jak3 led to a deficiency of these interactions, indicating negative controls (top and middle panels, second lane). Since Jak3 expression in the brain was unaffected in both Flox-Jak3 and Int-Jak3-KO mice fed with ND (Figure 3A), we determined the TREM2 interactions with Jak3 in the brains of these mice. Our data show that Jak3 interacted with TREM2 and vice versa in these mice (top and middle panels, third and fourth lanes). Feeding with HFD, however, led to a decrease in these interactions in both flox-Jak3 control and IEC-Jak3-KO mice (top and middle panels, fifth and sixth lanes). Next, we determined whether Jak3 association with TREM2 impacted the tyrosine phosphorylation of TREM2. The third panel from the top shows that TREM2 was tyrosine phosphorylated in WT, Flox-Jak3 (Flox) and Int-Jak3-KO (Vilcre) brains; however, feeding with HFD led to a decrease in tyrosine-phosphorylated TREM2. The bottom panel in Figure 4A shows the input controls for TREM2. Together, these results showed that Jak3 not only interacted with microglial TREM2, but HFD specifically affected Jak3 expression not only in the intestine but also in the brain, which led to a decrease in TREM2-co-immunoprecipitated Jak3 and decreased TREM2 tyrosine phosphorylation. Although the influences of the gut–brain axis on microglial functions have only been recently realized [54], the role of tyrosine kinases on such influences is not known. Since our data showed that Jak3 not only interacted with TREM2 but that Jak3 expression also suppressed β-Amyloid and pTau accumulations in the brain, we determined the extent to which Jak3 impacted microglial TREM2 functions, particularly TREM2 association with Aβ in the brain. The quantitation of Jak3′s impact on TREM2 as shown through the representative flow data from individual mice in Figure 4B and the corresponding statistical analysis from three independent experiments (five mice/group) in Figure 4B show that the global deficiency of Jak3 (KO) resulted in a two-fold decrease in TREM2 expression (WT-ND vs. KO-ND). Moreover, feeding with HFD resulted in about 2-fold decrease in TREM2 in WT mice, while Jak3-KO mice had an over 4-fold decrease in TREM2 expression. These results were further confirmed through immunofluorescence microscopy of the brain sections (Figure 4C) of WT and Jak3-KO mice under ND and HFD, which showed that a global deficiency of Jak3 (KO) resulted in a decrease in TREM2 expression (WT-ND vs. KO-ND), and feeding with HFD resulted in a decrease in TREM2 in WT mice, while Jak3-KO mice showed a further decrease in TREM2 expression. Moreover, the quantification of the florescent pixels from Figure 4C and additional (*n* = 5/group) representative images indicated similar folds of decrease in TREM2 expressions in the brain, which were similar to those obtained from the flow cytometric analyses in Figure 4B (Appendix A). To determine whether a global deficiency of Jak3 led decrease in TREM2 in the brain was due to IEC-specific Jak3, we used Int-Jak3-KO mice and their flox-Jak3 littermate controls. The flow cytometric data from individual mice in Figure 4D (left panel) and the corresponding statistical analyses from three independent experiments (five mice/group) in Figure 4D (right panel) show that a deficiency of IEC-Jak3 (Vilcre) resulted in a 30% decrease in TREM2 expression (FL-ND vs. Int-KO-ND). Moreover, feeding with HFD resulted in about 32% decrease in TREM2 in flox-controls (FL-ND vs. FL-HFD), while intestinal deficiency of Jak3 resulted in over 45% decrease in TREM2 expression (Int-KO -ND vs. Int-KO-HFD). These results were further confirmed through immunofluorescence microscopy of the brain sections (Figure 4E) of FL-control and Int-Jak3-KO mice under ND and HFD followed by quantifications of the florescent pixels (Appendix A), which showed that the IEC deficiency of Jak3 (Int-KO) resulted in a decrease in TREM2 expressions, which were further impacted by feeding with HFD, which resulted in a further decrease in TREM2 expressions (Figure 4E), and these results were similar to those obtained from the flow cytometric analyses.

Intestinal deficiency of Jak3 leads to increased microglial activation in the brain during HFD-induced obesity. Since Jak3-KO mice not only showed reduced TREM2-expression-associated increased Aβ accumulation in the brain, but these effects, in part, were also due to intestinal-specific deficiency of Jak3, we determined the impact of Jak3 on brain microglial activation. Ionized calcium binding adaptor molecule 1 (Iba1) is a microglia/macrophage-specific calcium-binding protein whose expression increases with microglial activation [55]. Figure 5A shows the flow cytometric data from individual mouse brains, and the corresponding statistical analyses from three independent experiments (five mice/group) are shown in Figure 5A (right bar graph). These data show that a global deficiency of Jak3 leads to a 3-fold increase in Iba1 positive cells in the brain (WT-ND vs. KO-ND), and while feeding with HFD leads to a similar increase in Iba1 positive cells in the WT mouse brain (WT-ND vs. WT-HFD), there were 3-fold and 10-fold increases in Iba1 positive cells in KO-mouse brain with HFD when compared to KO-ND and WT-ND mice, respectively. To determine whether the impact seen in the global deficiency of Jak3 (Jak3-KO) mice was due to IEC-specific Jak3, flow analyses were conducted using individual mouse brains from flox-Jak3-control (FL) and flox-Jak3-VilCre (Vilcre) mice in Figure 5B (left panel), and the corresponding statistical analyses from three independent experiments (five mice/group) in Figure 5B (right bar graph) show that the deficiency of IEC-Jak3 (Int-KO) resulted in a 2-fold increase in Iba1 positive cells (FL-ND vs. Int-KO-ND). Moreover, feeding with HFD resulted in about 3.5-fold increase in Iba1 positive cells in Flox-Jak3 (FL-ND vs. FL-HFD), and while feeding with HFD led to a 2.5-fold increase in Iba1 positive cells in the IEC-Jak3-KO mouse brain (Int-KO-ND vs. Int-KO-HFD), there was a 7-fold increase in Iba1 positive cells in the IEC-Jak3-KO brain with HFD when compared to WT-ND (WT-ND vs. Int-KO-HFD) mice. These results were further confirmed through immunofluorescence microscopy (Figure 5C) of the brain sections from FL-control and Int-Jak3-KO mice under ND and HFD followed by quantifications of the florescent pixels (Appendix A), which showed that the IEC deficiency of Jak3 (Int-KO) resulted in about 3-fold increase in Iba1 expressing cells (FL-ND vs. Int-KO-ND), and feeding with HFD resulted in about 5-fold increase in Iba1 positive cells in FL-control mice, while Int-Jak3-KO mice showed an over 10-fold increase in Iba1 positive cells in the brain.

Like Iba1, CD11b is expressed by activated microglia. It is involved in forming a part of complement receptor 3, which aids in the recognition and phagocytosis of antigens, including amyloid-β [56]. Next, we determined the impact of Jak3 on microglial activation, as determined through brain expression of CD11b. Figure 5D (left panel) shows the flow data for CD11b positive activated microglial cells from individual mouse brains, and the corresponding statistical analyses from three independent experiments (five mice/ group) are shown in Figure 5D (right panel). These data show that a deficiency of Jak3 leads to a 3-fold increase in CD11b positive microglial cells, and while feeding with HFD leads to an over 3-fold increase in these activated cells in the WT mouse brain, there is a 10-fold increase in these cells in the mouse brain with HFD compared to WT-ND mice. To determine whether these effects were due to IEC-Jak3, the flow data from individual mouse brains in Figure 5E (left panel) and the corresponding statistical analyses from three independent experiments (five mice/group) in Figure 5E (right bar graph) show that the deficiency of IEC-Jak3 resulted in a 2-fold increase in CD11b positive cells (FL-ND vs. Int-KO-ND). Moreover, feeding with HFD resulted in about 4-fold increase in these cells in Flox-Jak3 (FL-ND vs. FL-HFD), and the intestinal deficiency of Jak3 resulted in an over 6-fold increase in CD11b positive cells in the brain compared to Flox-Jak3 mice (FL-ND vs. Int-KO-HFD). These results were further confirmed through immunofluorescence microscopy (Figure 5F) of the brain sections from FL-control and Int-Jak3-KO mice under ND and HFD followed by quantifications of the florescent pixels (Appendix A), which showed that the IEC deficiency of Jak3 resulted in increased CD11b positive cells, and feeding with HFD resulted in about 4-fold increase in CD11b positive cells in FL-control mice, while Int-Jak3-KO mice showed an over 8-fold increase in CD11b positive cells in the brain.

EGF-like module containing mucin-like hormone receptor-like 1 (EMR1), also known as F4/80, is a member of the adhesion GPCR family of proteins [57]. Although human EMR1 expression is restricted to eosinophils [58], the murine homolog F4/80 is widely used as a marker for murine tissue macrophage populations [59]. Therefore, we determined whether the expression of Jak3 had an impact on the brain expression of F4/80 as an indicator of microglial activation. Appendix A shows the flow data for F4/80 positive activated microglial cells from individual mouse brains, and the corresponding statistical analyses from three independent experiments (five mice/ group) are shown in Appendix A. These figures show that the deficiency of Jak3 (KO) led to a 2-fold increase in F4/80 positive activated microglial cells in the brain, and while feeding with HFD led to a similar increase in the activated microglial cells in the WT mouse brain, there was a 4-fold increase in F4/80 positive microglial cells in mouse brains with HFD (WT-ND vs. KO-HFD). To determine the tissue-specific impact of IEC-Jak3 on microglial activation in the brain, the flow data from individual mouse brains in Appendix A and the corresponding statistical analyses from three independent experiments (five mice/ group) in Appendix A show that the deficiency of IEC-Jak3 (Int-KO) resulted in a 3-fold increase in F4/80 positive cells (FL-ND vs. Int-KO-ND). Moreover, HFD promoted about 2.5-fold increase in F4/80 positive cells in Flox-control (FL-ND vs. FL-HFD), while the intestinal deficiency of Jak3 led to an over 20-fold increase in F4/80 positive cells in the brain (FL-ND vs. Int-KO-HFD). These results were further confirmed through immunofluorescence (Appendix A) microscopy of the corresponding mouse brains followed by quantifications of the florescent pixels (Appendix A), which showed that the IEC deficiency of Jak3 (Int-KO) resulted in about 4-fold increase in F4/80 positive cells (FL-ND vs. Int-KO-ND), and feeding with HFD resulted in about 5-fold increase in F4/80 positive cells in FL-HFD mice, while Int-Jak3-KO mice showed an over 10-fold increase in F4/80 positive cells in the brain (FL-ND vs. Int-KO-ND).

HFD-led suppression of Jak3 expression in the brain is responsible for microglial activation through increased TLR expression mediated brain inflammation. It is reported that although F4/80 may not be essential for the development of tissue-specific macrophages, it is involved in the induction of regulatory T-cell-mediated tolerance [60]. Since our data showed a deficiency of IEC-Jak3 led to a decreased microglial TREM2 and increased microglial activation, as indicated by increased CD11b and F4/80, we determined whether these effects were due to central TLRs. Figure 6A shows the flow data for TLR-4 positive cells from individual mouse brains and the corresponding statistical analyses from three independent experiments (five mice/group). These data show that global Jak3 deficiency (KO) led to a 3-fold increase in TLR-4 positive cells in the brain of mice fed with ND, and feeding with HFD led to a similar increase in TLR-4 positive cells in the WT mouse brains. However, there was a 4-fold increase in TLR-4 positive cells in mouse brains with HFD in the group of Jak3-KO mice (WT-ND vs. KO-HFD). To determine the impact of IEC-Jak3 on TLR-4 expression in the brain, we first performed a western analysis of Jak3 expression in the brains of IEC-Jak3-KO mice. Since we showed that flox-Jak3 (F/F) mouse brains not only expressed Jak3, but IEC deficiency of Jak3 (Vilcre/F/F) had no impact on Jak3 expression in the brains, we determined the impact of intestinal deficiency of Jak3 on TLR-4 expression in the brain. The flow data from individual mouse brains in Figure 6B and the corresponding statistical analyses from three independent experiments (five mice/ group) in Figure 6B (right bar graph) show that the deficiency of IEC-Jak3 (Int-KO) resulted in a 5-fold increase in TLR-4 positive cells (FL-ND vs. Int-KO-ND) in mice fed with ND. Moreover, HFD promoted an approximately similar increase in TLR-4 positive cells in Flox-Jak3 (FL-ND vs. FL-HFD). However, the intestinal deficiency of Jak3 led to an over ten-fold increase in TLR-4-positive cells in the brain of mice fed with HDF (FL-ND vs. Int-KO-HFD). These results were further confirmed through immunofluorescence imaging of the corresponding mouse brains (Figure 6C) followed by quantifications of the florescent pixels (not shown), indicating that the IEC deficiency of Jak3 (Int-KO) resulted in about 5-fold increase in TLR-4 positive cells (FL-ND vs. Int-KO-ND), and feeding with HFD resulted in about 5-fold increase in TLR-4 positive cells in FL-HFD mice, while Int-Jak3-KO mice showed an increase in TLR-4 positive cells in the brain. Microglia transform from a resting stage into activated stages, characterized by changes in the morphology and expression of cytoplasmic and surface proteins [61]. Since our data showed that a deficiency of Jak3 expression led to the activation of microglia, as characterized by the increased expression of CD11b and F4/80, we determined whether this led to brain inflammation. The determinations of pro-inflammatory cytokines TNF-α and IL-6 using the flow data from individual mouse brains in Figure 6D and the corresponding statistical analyses from three independent experiments (five mice/ group) in Figure 6D (right bar graph) show that the global deficiency of Jak3 (KO) resulted in a 10-fold increase in TNF-α positive cells (WT vs. KO) in mice fed with ND. Moreover, HFD promoted about 9-fold increase in TNF-α positive cells in WT (WT-ND vs. WT-HFD) mice, and a global deficiency of Jak3 led to an over 28-fold increase in TNF-α positive cells in the brain when compared with WT mice fed with ND (WT-ND vs. KO-HFD). These results were further confirmed through the determination of another pro-inflammatory cytokine IL-6 (Figure 6E), which showed that the global deficiency of Jak3 (KO) resulted in a 10-fold increase in IL-6 positive cells (WT vs. KO) under ND. Moreover, HFD promoted an approximately similar increase in IL-6 positive cells in WT mice when compared to WT fed with ND (WT-ND vs. WT-HFD). Moreover, the global deficiency of Jak3 led to an over 18-fold increase in IL-6 positive cells in the brain of mice fed with HFD when compared to WT fed with ND (WT-ND vs. KO-HFD). To determine whether the microglial activated brain inflammations were due to intestinal epithelial Jak3, we determined TNF-α (Figure 6F) and IL-6 (Figure 6G) using the flow data from individual IEC-Jak3-KO mouse brains and the corresponding statistical analyses from three independent experiments (five mice/group) in Figure 6F,G (right panel bar graphs). Our data show that the IEC deficiency of Jak3 (Int-KO) resulted in a 5-fold increase in TNF-α positive cells (Flox-ND vs. Int-KO-ND) in mice fed with ND. Moreover, feeding with HFD promoted about 6-fold increase in TNF-α positive cells in Flox-Jak3 (Flox-ND vs. Flox-HFD), and IEC deficiency of Jak3 (Int-KO) led to an over 19-fold increase in TNF-α positive cells in the brain (Flox-ND vs. Int-KO-HFD). These results were further confirmed through the determination of another pro-inflammatory cytokine IL-6 (Figure 6G), which showed that either IEC deficiency of Jak3 (Int-KO) alone or when fed with HFD both resulted in increased IL-6 positive cells in the brain (Figure 6G, right bar graph).

Jak3 deficiency, either by genetic manipulation or HFD, leads to increased HIF1α and decreased microglial TREM-2-expression-associated increased Abeta accumulation in the brain. Microglia is activated by several pro-inflammatory factors, including oxidative stress, creating hypoxic conditions [62,63]. Since our data showed that the deficiency of Jak3 not only activated brain microglia but also led to increased pro-inflammatory cytokines in the brain, we determined whether these effects were due to the transcription factor HIF-1α, induced by hypoxia, known for activating pro-inflammatory cytokines in tissue-specific macrophages [64]. Figure 7A shows the flow data from individual Jak3-KO mouse brains, and the corresponding statistical analyses from three independent (five mice/ group) experiments are shown in Figure 7A (right panel). These data indicate that Jak3 deficiency results in a 5-fold increase in HIF1-α positive cells (WT-ND vs. KO-ND) under ND. Moreover, although HFD promoted a similar increase in HIF1-α positive cells in WT mice (WT-ND vs. WT-HFD), Jak3 deficiency caused an over 10-fold increase in HIF1-α positive cells in the brain with HFD when compared to wild-type mice (WT-ND vs. KO-HFD). To determine whether intestinal Jak3 had an impact on HIF1-α in the brain, the flow data from individual IEC-specific Jak3-KO mouse brains in Figure 7B and the corresponding statistical analyses from three independent experiments (five mice/ group) in Figure 7B (right panel) show that IEC deficiency of Jak3 (Int-KO) resulted in an over 2.5-fold increase in HIF1-α positive cells under ND (Flox-ND vs. Int-KO-ND). Although HFD induced an over 4-fold increase in HIF1-α positive cells in Flox-Jak3 (Flox-ND vs. Flox-HFD), IEC deficiency of Jak3 (Int-KO) resulted in an over 5-fold increase in HIF1-α positive cells in the brain (WT-ND vs. Int-KO-HFD). Since our data showed that Jak3 interacts with TREM2, and a deficiency of Jak3 leads not only to reduced tyrosine phosphorylation of TREM2 but also reduced expression of TREM2 in the brain, we therefore investigated the impact of Jak3 on TREM2-mediated microglial clearance of β-amyloid. To achieve that, using flowcytometry, we first determined the impact of Jak3 and HFD on TREM2 positive cells in the brain followed by a determination of β-amyloid-TREM2 double positive cells in the same tissue samples using flowcytometry and immunofluorescence microscopy as a measure of compromised clearance of β-amyloid plaques. Figure 7C shows the flow data from individual mouse brains, and the corresponding statistical analyses from three independent experiments (five mice/group) are shown in the lower bar graphs. The data shown on the top panels and the bottom left indicate that under ND, Jak3 deficiency led to an over five-fold increase in β-amyloid positive cells in the brain (WT-ND vs. KO-ND). Moreover, although feeding with HFD led to a 3-fold increase in β-amyloid positive cells in WT mice, a deficiency of Jak3 led to an over 14-fold increase in β-amyloid positive cells with HFD compared to WT-ND. Since the deficiency of Jak3 led to increased β-amyloid deposition, and TREM2 is a receptor for β-Amyloid that mediates the microglial functions, we investigated the reason for increased β-amyloid deposition in the absence of Jak3. Figure 7C (bottom middle graph) shows that the deficiency of Jak3 led to an over three-fold decrease in TREM-2 expression in mouse brains, where HFD had similar effects in WT mice (WT-HFD). Moreover, the deficiency of Jak3 led to an over five-fold decrease in TREM-2 positive cells in mouse brains with HFD. Figure 7C (bottom right graph) shows the statistical analysis of β-amyloid and TREM-2 double positive cells, which shows that both KO-ND and WT-HFD mice had increases in β-amyloid and TREM-2 double positive cells compared to WT-ND mice, where Jak3-deficient mice had a slight increase in these cells compared to WT-HFD and a significant increase compared to WT-ND mice. Moreover, Jak3-deficient mice fed with HFD had the least double positive cells, the reason for which is unclear. Together, these results show that the deficiency of Jak3, either by genetic manipulation or with HFD, led to a decreased expression of β-amyloid receptor TREM-2, which led to a decreased uptake of β-amyloid by microglial cells in the brain, resulting in an increased deposition of β-amyloid. Since our data show increased pro-inflammatory microglia, as indicated by CD11b positive cells, and decreased anti-inflammatory microglia, as indicated by decreased TREM-2, indicating M1 polarization of the brain microglial cells, we determined whether this polarization impacted the microglial deposition of β-amyloid. Figure 7D shows the flow data from individual mouse brains, and the corresponding statistical analyses from three independent experiments (five mice/group) are shown in Figure 7D (lower panels). These data show that Jak3 deficiency, either by genetic manipulation in KO mice or with HFD-mediated suppression of Jak3 expression in WT mice, led to a significant increase in not only the deposition of β-amyloid in the brain (Figure 3) but also in β-amyloid positive microglial cells, as indicated by β-amyloid and Iba1 double positive cells in the brain (Figure 7D, lower left and right panels, respectively).

The intestinal epithelial deficiency of Jak3 is responsible for reduced TREM2-associated β-Amyloid accumulation in the brain. Since our data showed that the total body deficiency of Jak3 led to decreased TREM-2 expression in microglial cells and increased Abeta accumulation in the brain, we determined whether the intestinal epithelial tissue Jak3 had an impact on reduced TREM2-associated β-Amyloid deposition. Figure 7E (upper panels) shows the flow data from individual mouse brains with intestinal deficiency of Jak3 expression and the corresponding statistical analyses from three independent experiments (five mice/group) (Figure 7E, lower panels). Figure 7E shows that the intestinal epithelial deficiency of Jak3 led to an over 5-fold increase in β-amyloid positive cells in the brain (Flox-ND vs. Int-KO-ND). Moreover, although mice fed with HFD had a similar increase in β-amyloid positive cells to the Flox mice (FL-HFD), the intestinal deficiency of Jak3 led to an over 12-fold increase in β-amyloid positive cells with HFD compared to Flox-ND. Since the global deficiency of Jak3 led to a decrease in TREM2 in the brain, we investigated whether the intestinal Jak3 had an impact on decreased TREM2. Figure 7E (lower middle panel) shows that either the intestinal deficiency of Jak3 and/or HFD led to an over three-fold decrease in TREM-2 expression in mouse brains. Figure 7E (lower right panel) shows the statistical analysis of β-amyloid and TREM-2 double positive cells, which shows that there was no significant change in the double positive cells in the four groups. Next, we determined the impact of the intestinal epithelial Jak3 on microglial activation in the brain using microglial Iba1 expression as a marker. Figure 7F shows the flow data from individual mouse brains with the intestinal deficiency of Jak3 expression and the corresponding statistical analyses from three independent experiments (five mice/group) (Figure 7F, lower panels). These results show that the intestinal epithelial deficiency of Jak3 and/or feeding with HFD led not only to a significant increase in β-amyloid accumulation in the brain (Figure 7F, lower left panel), but these increases were also associated with a significant increase in microglial activation, as indicated by β-amyloid Iba1 double positive cells (Figure 7F, lower right panel), indicating Jak3 deficiency led to an increased number of activated and β-amyloid positive microglial cells.

Jak3 deficiency promotes pTau accumulation through decreased TREM-2 expression in microglial cells. We investigated the impact of Jak3 on TREM2-mediated microglial clearance of pTau. To achieve that, using flowcytometry, we first determined the impact of Jak3 and/or HFD on TREM2 positive cells in the brain followed by a determination of pTau-TREM2 double positive cells in the same tissue samples using flowcytometry and immunofluorescence microscopy as a measure of compromised clearance of pTau. Figure 8A (top panels) shows the flow data from individual mouse brains and the corresponding statistical analyses from three independent experiments (five mice/ group) (Figure 8A, lower panels). Figure 8A (lower left panel) shows that under ND, Jak3 deficiency led to an over 3-fold increase in pTau positive cells in the brain (WT-ND vs. KO-ND). Moreover, although feeding with HFD led to a 2-fold increase in pTau positive cells in WT mice, Jak3 deficiency caused an over 10-fold increase in pTau positive cells with HFD compared to WT-ND. Since TREM2 is a receptor for pTau that mediates microglial phagocytosis, we investigated the reason for increased pTau deposition in the absence of Jak3. Figure 8A (lower middle panel) shows that the deficiency of Jak3 led to a 10-fold decrease in TREM2 positive cells in mouse brains, where HFD caused a 3-fold decrease in TREM2 in WT mice (WT-HFD). Moreover, the deficiency of Jak3 and HFD together caused an 11-fold decrease in TREM-2 positive cells in mouse brains (WT-ND vs. KO-HFD). Figure 8A (lower right panel) shows the statistical analysis of pTau and TREM-2 double positive cells in mouse brains, which indicates that although there was a non-statistical increase in double positive cells in KO-ND compared to WT-ND, feeding with HFD led to an over two-fold increase in double positive cells in mouse brains compared to WT-ND mice (WT-ND vs. WT-HFD). Moreover, Jak3 deficiency combined with HFD led to a decrease in these double positive cells, which were comparable to WT-ND mice. Together, these results show that Jak3 deficiency, either by genetic manipulation or with HFD, led to a decreased expression of pTau receptor TREM-2, decreased uptake of pTau by TREM2 positive microglial cells and increased deposition of pTau in the mouse brains. Since our data show M1 polarization of microglial cells in the brain by the total body Jak3 deficiency, we determined whether this polarization was impacted by the intestinal epithelial tissue-specific Jak3. Figure 8B shows the flow data from individual mouse brains with the intestinal epithelial deficiency of Jak3 expression, and the corresponding statistical analyses from three independent experiments (five mice/group) are shown in Figure 8B (lower panels). Figure 8B (lower left panel) shows that the intestinal epithelial deficiency of Jak3 led to a 1.3-fold increase in pTau positive cells in mouse brains when they were fed with ND (Flox-ND vs. Int-KO-ND). Moreover, although feeding with HFD led to a 2-fold increase in pTau positive cells in Flox mice (FL-ND vs. FL-HFD), the intestinal epithelial deficiency of Jak3 combined with HFD led to an over 2.6-fold increase in pTau positive cells compared to Flox-ND groups of mice. Since the global deficiency of Jak3 led to a decrease in TREM2 in the brain, we further investigated whether the intestinal epithelial specific deficiency of Jak3 had an impact on such decrease. Figure 8B (lower middle panel) shows that the intestinal epithelial deficiency of Jak3 led to a 1.8-fold decrease in TREM-2 expression in mouse brains. Moreover, feeding with HFD alone (Flox-ND vs. Flox-HFD) or combined with the intestinal epithelial deficiency of Jak3 (Flox-ND vs. Int-KO-HFD) led to a 7-fold decrease in TREM-2 positive cells in mouse brains. Figure 8B (lower right panel) shows the statistical analyses of pTau and TREM-2 double positive cells, which indicate that there was no statistical difference in double positive cells among the FL-ND, FL-HFD, Int-KO and Int-KO-HFD groups of mice. Next, we determined the impact of intestinal epithelial Jak3 on microglial activation in the brain using F4/80 as a marker for microglial activation. The immunofluorescence microscopy using mouse brain sections and F4/80, pTau antibodies and DAPI for nuclei in Figure 8C shows that either HFD alone and/or intestinal epithelial deficiency of Jak3 led to increased F4/80 positive microglial cells, and these increases were associated with increased pTau accumulation in mouse brain sections. We also corroborated these results using the flow cytometry from individual mouse brains with either whole-body deficiency (Figure 8D) or intestinal epithelial tissue-specific deficiency (Figure 8E) of Jak3 expression and the corresponding statistical analyses from three independent experiments (five mice/group) in the right panels of the corresponding figures. These data show that the global deficiency of Jak3 (Figure 8D) led to a two-fold increase in F4/80 positive cells (WT-ND vs. KO-ND), which were similar to the increase in these cells with HFD-fed WT mice group (WT-ND vs. WT-HFD). However, feeding Jak3-deficient mice with HFD led to a four-fold increase in F4/80 positive cells in the brain (WT-ND vs. KO-HFD). To determine the impact of the intestinal epithelial tissue-specific Jak3 deficiency on F4/80 positive microglial cells in the brain, Figure 8E shows that the intestinal epithelial deficiency of Jak3 led to a three-fold increase in F4/80 positive cells (FL-ND vs. Int-KO-ND), which was statistically similar to the increase in these cells in the HFD-fed flox mice group (FL-ND vs. FL-HFD). However, feeding intestinal epithelial Jak3-deficient mice with HFD led to a 15-fold increase in F4/80 positive cells in the brain (Flox-ND vs. Int-KO-HFD). These data indicate that the intestinal epithelial Jak3 deficiency had a significant impact on microglial activation in the brain and its phagocytic functions toward Abeta and pTAu.

The intestinal epithelial deficiency of Jak3 is responsible for systemic-inflammation-led reduced Iba1 expression and compromised-microglial-function-led Abeta and pTau accumulation in the brain. Figure 9A (top panels) shows the flow data for pTau and microglial marker Iba1 double positive cells from individual mouse brains, and the corresponding statistical analyses from three independent experiments (five mice/group) are shown in Figure 9A (bottom left panel). These data show that the deficiency of Jak3 leads to a 2.5-fold increase in pTau and Iba1 double positive cells in mouse brains, and while feeding with HFD leads to a similar increase in these double positive cells in the WT mouse brains, there was a 7-fold increase in pTau and Iba1 double positive cells in mouse brains that were deficient in Jak3 and were fed with HFD when compared to WT-ND mice. To determine the impact of Jak3 deficiency on pTau positive cells, Figure 9A (bottom right panel) shows similar trends of pTau deposition in the brain, where the deficiency in Jak3 led to a five-fold increase in pTau positive cells, which was similar to those of WT fed with HFD. However, there was a 15-fold increase in pTau positive cells in mouse brains that were deficient in Jak3 and were fed with HFD when compared to WT-ND mice (WT-ND vs. KO-HFD). To determine the impact of intestinal epithelial deficiency of Jak3 on pTau and Iba1 double positive cells in mouse brains, the flow data from individual mouse brains in Figure 9B (top panels) and the corresponding statistical analyses from three independent experiments (five mice/ group) in Figure 9B (bottom right panel) show that the deficiency of IEC-Jak3 (Int-KO) resulted in a 2-fold increase in Iba1 and pTau double positive cells (FL-ND vs. Int-KO-ND). Moreover, feeding with HFD resulted in about 2.5-fold increase in pTau and Iba1 double positive cells in Jak3-Flox mice (FL-ND vs. FL-HFD). Moreover, the intestinal epithelial deficiency of Jak3 combined with feeding with HFD resulted in about 5-fold increase in Iba1 and pTau double positive cells in the brain (WT-ND vs. Int-KO-HFD). To determine the impact of the intestinal epithelial deficiency of Jak3 on pTau positive cells in mouse brains, Figure 9B (bottom right panel) shows similar trends of pTau deposition in the brain, where the intestinal epithelial deficiency in Jak3 led to a six-fold increase in pTau positive cells, which was lower than those of flox mice fed with HFD. Moreover, there was a 10-fold increase in pTau positive cells in mouse brains that were deficient in intestinal epithelial Jak3 and were fed with HFD when compared to FL-ND mice (FL-ND vs. Int-KO-HFD). Finally, we determined the colocalization of pTau and Iba1 in the immunofluorescence brain sections of the Flox-HFD and Int-Jak3-KO-HFD mice in Figure 9C followed by intracellular quantifications of the intensities of the colocalized florescent pixels (Figure 9C, bottom panel). These results show a significant decrease in intracellular intensities of the colocalized florescent pixels for pTau and Iba1 in Int-Jak3-KO-HFD mouse brains compared to Flox-HFD brains, indicating an impairment of pTau clearance due to the intestinal epithelial deficiency of Jak3.

## 3. Discussion

Obesity comes with an increased risk of neurodegenerative diseases [65], including dementia [66,67]. Meta-analyses show strong associations between obesity, AD and other forms of dementias, where obesity doubles the risk of AD [67,68]. Postmortem studies on cases of elderly with morbid obesity show increased concentrations of β-amyloid and tau proteins, the pathology markers associated with AD [69]. Since cognitive impairment is now recognized as a major co-morbidity of obesity [70,71], we investigated the mechanism of the nutritional driver for obesity on cognitive functions. Previously, we reported that obese humans have a reduced expression of intestinal Jak3 46, and a chronic reconstitution of these conditions in our novel mouse model led to obesity-associated metabolic syndrome [21]. To determine the mechanism, our in vivo data in this study show that the ingestion of HFD leads to a decrease in the intestinal mucosal expression of Jak3, and these decreased expressions of Jak3 are associated with increased expressions of TLR-2 and TLR-4 in intestinal mucosal tissues. These results corroborate our previous report, where we showed that Jak3 suppresses the intestinal mucosal TLR expression through the AKT pathways. Here, we showed that HFD-led mucosal Jak3 deficiency leads to increased TLR expression and the corresponding increase in phosphorylated AKT (pAKT) and phosphorylated NFκB (pNFκB) (Figure 1A,B) in WT mice.

Accumulating evidence in the literature suggests a strong relationship between diet and health, where, among others, the gut microbiota is a key influencing factor [72]. For example, in mice, feeding with HFD leads to increases in the abundance of Firmicutes, Proteobacteria and Actinobacteria, and concomitant reductions in health-promoting bacteria, such as those from the Bacteroidetes phylum, and Bifidobacterium and Akkermansia genera [73,74,75,76]. Dietary components can shape the gut microbiota, where studies also indicate HFD as promoting gut dysbiosis in both animals and humans [77]. However, the information on the signaling pathways and the molecular mechanisms are poorly understood. Since our study showed HFD mediate intestinal Jak3-deficiency-led increase in TLR expression, we reconstituted these conditions using Jak3-KO mice and determined the impact Jak3 deficiency had on the changes in the gut microbiome composition. To avoid the influences of cage-related variation in the microbiome and seeding from the maternal microbiome, we used samples from co-housed WT and Jak3-KO littermates, where the only differences were the expressions of Jak3. As expected, the time courses of the changes in the gut microbiome showed that, at the time of weaning, the relative abundances of Bacteroidetes and Firmicutes were similar or within the margin of error. However, as the mice grew older, although there was a significant shift in these abundances in the gut microbiome of both WT and Jak3-KO co-housed mice, Jak3-KO mice showed a significant increase in Firmicutes and a corresponding decrease in Bacteroidetes. Studies suggest increases in Firmicutes are associated with gut inflammation [78], where Clostridiales are among the significantly changed orders. Our data suggested that intestinal Jak3 may be involved in such a shift. This is because Jak3 deficiency resulted in a 3-fold increase in the class Clostridia, order Clostridiales. Moreover, Jak3-KO mice also had about 3-fold increase in Lachnospiraceae and an over 2-fold increase in Ruminococcaceae families. As our previous report suggested that global Jak3 deficiency leads to obesity-associated metabolic syndrome [21], this study demonstrated that global Jak3 deficiency in mice also leads to colonic dysbiosis. Studies suggest gut dysbiosis also facilitates CLGI-associated neurodegeneration [10,50], although the mechanisms are poorly understood.

HFD is the main driver of diabetes and obesity worldwide, where more than 35% of people in the US are obese, and 9% of adults have diabetes [79]. Reports suggest both obesity [8,9] and glycemic dysregulations in type 2 diabetes [80,81] as key morbidities that occur before the onset of cognitive impairment, although the molecular mechanisms are far from clear. Since our data suggested the key role of Jak3 in both obesity-associated metabolic syndrome and gut dysbiosis, our natural next step was to see whether these mice with impaired Jak3 signaling had any cognitive functions impacted. Our data suggested the mice with impaired Jak3 signaling not only had a significant increase in cognitive impairment parameters, particularly related to working memory and roaming memory, but these increases were also associated with increased deposition of β-amyloid and pTau in the brain. Interestingly, our data suggested that mice fed with HFD had increased accumulation of β-amyloid and pTau in the brain during chronic Jak3 deficiency. This led us to investigate whether mouse brains expressed Jak3. Our data showed that mouse brains not only expressed Jak3, but feeding with HFD caused a moderate decrease in Jak3 expression in the brain in WT mice, whereas chronic Jak3 deficiency in mice led to significant increases in the accumulation of β-amyloid and pTau compared to WT mice (Figure 1). An impairment in the microbiota–gut–brain (MGB) axis has been suggested in several neurodegenerative diseases with cognitive dysfunction [82,83], where the circulating gut microbial products have been implicated in a causative role. However, the roles of non-receptor tyrosine kinase in general and Jak3 in particular in the gut-dysbiosis-mediated cognitive impairment were not known. Our data showed that HFD not only causes Jak3 deficiency in the gut, but it also impacts Jak3 expression in the brain, and the recapitulation of chronic Jak3 deficiency using global Jak3-KO mice causes spatial cognitive impairment (Figure 1). Interestingly, both global and intestinal Jak3-deficient mice also had a 5-fold increase in plasma LPS (unpublished observation).

The molecular understanding of the instigator of obesity-associated metabolic syndrome and the related comorbidities in the human body is unknown. Since our previous report suggested that global Jak3 deficiency in mice leads to obesity-associated type 2 diabetes, in this study, we showed the same global Jak3 deficiency also causes gut-dysbiosis-associated cognitive impairment. To demonstrate which tissue-specific Jak3 deficiency instigated such conditions, our data suggested metabolic-syndrome-associated cognitive impairment, as seen in global Jak3-deficient mice, due to intestinal epithelial cell-specific Jak3 deficiencies, where IEC-Jak3-KO mice showed significant increases in body weight with HFD and compromises in restoring blood glucose upon glycemic challenge (Figure 2). Although obesity-associated metabolic syndrome comes with a cognitive impact, the intestinal mucosal specific role of Jak3 in such comorbidities was not known. Our data showed that although, over the sessions, the flox control counterparts of IEC-Jak3-KO animals had retained the spatial cognition task, the IEC-specific deficiency of Jak3 nevertheless predisposed the animals not only toward HFD-led obesity-associated glycemic dysregulation but also to its comorbidity of significant deficiency in spatial cognition, particularly with respect to working memory, roaming memory, reward memory and errors committed before reaching a particular destination. Our data suggested that WT and flox-Jak3 controls and IEC-Jak3-KO mice expressed Jak3 in the brain, and Jak3 expression remained unaffected under a normal diet, indicating that the impacts on mid-age spatial cognition impairment, as seen with the global Jak3-deficient mice under a normal diet, were due to intestinal specific deficiency of Jak3. This was because, while normal diet did not significantly affect either Jak3 expression in the brain or the associated β-amyloid or pTau accumulation in the brain, high-fat diet specifically affected Jak3 expression in the brain only when there was a deficiency in IEC-Jak3. Moreover, the HFD-led impacts on Jak3 in the brain in IEC-Jak3-deficient mice were also associated with increased accumulation of β-amyloid or pTau in the brain, and indeed, these were further reflected in the deficiency of spatial cognition in these mice (Figure 3). Together, these data proved that the deficiency of Jak3 led impacts on cognition were due to the accumulation of Abeta and pTAu in the brain, which were again due to the IEC-specific deficiency of Jak3. Since, previously, we reported that Jak3 facilitated intestinal epithelial wound repair [30] and mucosal barrier functions [20], the current study showed that Jak3′s intestinal functions are essential for protection against not only obesity-associated glycemic dysregulation but also against its comorbidity of cognitive impairment.

To determine the mechanism of how intestinal Jak3 regulated the accumulation of Abeta and pTau in the brain, we investigated the microglial functions of the brain in mice with either global or intestinal deficiency of Jak3. First identified in dendritic cells and macrophages, the triggering receptor expressed on myeloid cells 2 (TREM2) is a single-pass transmembrane receptor on the microglial cells in the brain [84]. TREM2 functions are essential for phosphorylated Abeta sensing and its subsequent clearance by microglial cells in the brain, and a compromise in these functions has been associated with cognitive impairments [85]. Since our data suggested for the first time that mouse brains expressed Jak3, we investigated whether Jak3 interacts with TREM2 in mouse brains and showed that Jak3 not only interacts with TREM-2, but a global Jak3 deficiency in mice leads to an impairment of these interactions and the associated tyrosine phosphorylation of TREM2. Surprisingly, however, IEC-specific deficiency of Jak3 impaired these interactions in the brain and impacted tyrosine phosphorylation of TREM2, particularly in HFD-fed mice, indicating that the TREM2–Jak3 interactions in the brain were regulated by Jak3 functions in the gut (Figure 4). Moreover, these results were also corroborated by respective decreases in TREM2 positive cells and corresponding increases in Iba1 and CD11b positive cells in the brain tissues of these mice, indicating increased microglial activation because of deficient Jak3 interactions with TREM2. We speculate that this could be due to HFD-led impact on the suppression of Jak3 expression, as shown in the IEC [46].

To determine how IEC deficiency of Jak3 led to the activation of microglial cells in the brains of HFD-fed mice, our data showed that a global deficiency of Jak3 promotes an increase in TLR4 positive cells in the brain, and these conditions were reconstituted by HFD-fed mice under IEC deficiency of Jak3 (Figure 6). As HFD directly interacts with the gut as opposed to the brain, IEC deficiency of Jak3 led to increased TLR activation in the gut, which ultimately impacted the brain through the TLR agonist (Figure 10). Since the downstream targets of TLR activation are the proinflammatory cytokines, both of these conditions of increased TLR4 positive cells impacted the brain inflammation, as indicated by a higher level of IL-6 and TNF-α. Studies suggest that peripheral inflammation has a significant impact on central inflammation, with different pro-inflammatory cytokines leading to neuroinflammation [86]. Our previous study indicated a global deficiency of Jak3 leads to systemic CLGI, where intestinal Jak3 promoted tolerogenic effects through suppressing TLR signaling [21]. The current study extended those findings by proving an essential role of IEC-Jak3 in cognitive impairment, where impairment of intestinal Jak3 signaling acted as the source of brain inflammation through increased TLR signaling. Moreover, this also indicates a redundancy in Jak3-mediated tolerogenic impact between the intestinal epithelial and microglial cells, where HFD promotes Jak3 suppression mediated activation of TLR signaling and the associated inflammation. A meta-analysis of 175 studies indicated an elevated level of peripheral IL-6, TNF-α and decreased IL-1 receptor antagonist and leptin in patients with AD compared with healthy controls, where IL-6 levels inversely correlated with mean Mini-Mental State Exam / Folstein Test scores [87]. Although it was reported that microglia TREM2 expression in vivo and in vitro sharply declined in response to acute pro-inflammatory stimuli [15,88,89], the underlying mechanisms were, nevertheless, less understood. Our data show that Jak3 promotes mucosal tolerance in the intestine through IEC-Jak3 and central tolerance through microglial Jak3 by suppressing the intestinal-originated central inflammation by way of suppressing TLR-mediated signaling. In addition, our data suggested that Jak3 also promotes microglial TREM2 expression and plaque and tangle clearance. This could be through Jak3-mediated TREM2 phosphorylation, its stabilization and phagocytosis of the plaque and tangle proteins.

Histologically, dementia is characterized by an abundance of Aβ plaques in the brain and an increased presence of the hyperphosphorylated form of microtubule-associated protein tau (pTau) aggregates within the neurons [90]. One of the histological features of dementia is the presence and accumulation of reactive microglial cells around Aβ plaques. Microglial involvement in dementia pathogenesis was revealed through a discovery that a rare variant of the gene encoding TREM2 confers several-fold increased risk of AD in humans [90,91]. TREM proteins participate in diverse cell processes, including inflammation, bone homeostasis, neurological development and coagulation. TREM2 is a member of the TREM family, which includes TREM1, TREML1 (TREM-like 1), TREML2 and TREML4. TREM2 is highly and exclusively expressed on the cell surface of brain microglia [52,53]. As a microglia surface receptor, TREM2-mediated signaling promotes phagocytosis and microglial survival [92,93,94]. Studies also suggest TREM-2 functions in inhibiting cytokine production by macrophages in response to the TLR activation [95]. For example, bone-marrow-derived macrophages isolated from Trem2-deficient mice released more inflammatory cytokines (TNFα and IL-6) upon stimulation with TLR agonists (LPS, CpG and zymosan) compared to WT cells [96], suggesting that TREM2 signaling might play an anti-inflammatory role by inhibiting the TLR pathway [15]. Our data showed the symptoms of Abeta plaque and pTau accumulation were significantly increased in global Jak3-deficient mice, and IEC-specific deficiency of Jak3 recapitulated these conditions, particularly under HFD. Moreover, these symptoms could be due to decreased TREM2 positive cells, decreased microglial survival and compromised microglial phagocytosis of the plaque and tables because of Jak3 deficiency (Figure 7). This was also because TREM2 is implicated in phagocytosis; a knockdown of TREM2 in mice reduced phagocytosis by microglia [94]. Moreover, our data suggested that brain deficiency of Jak3 could be instigated by the deficiency in intestinal Jak3 signaling, particularly under HFD. Oligomeric Aβ and tangled phosphorylated tau are known to stimulate microglial cytokine and chemokine production while decreasing their phagocytic capacity during AD [97,98]. Studies suggest that microglia exposed to hypoxic conditions activate TLR4 and produce TNF-α [99] in a HIF-1-dependent mechanism. The present study suggested that feeding with HFD under the intestinal deficiency of Jak3 caused brain induction of hypoxia inducible factor-1 as a mechanism of TLR4-activation-led brain inflammation.

Tau is an axonal and highly soluble protein. It associates with the microtubules present in the neurons of the central nervous system. Tau forms neurofibrillary tangles as a tau pathology found in AD/dementia. Moreover, microglial activation is shown to precede neurofibrillary tangle formation in tau transgenic mice [100]. The role of intestinal Jak3 in tauopathies was not known. Our data showed that IEC deficiency of Jak3 results in microglial activation in the brain through decreased TREM and increased TLR4 signaling associated increased pro-inflammatory cytokines in the brain. Moreover, these effects were associated with increased pTau deposits. We speculate that the deficiency of IEC-Jak3 led gut dysbiosis might be contributing to microglial activation, which could be preceding neurofibrillary tangle formation by tau in IEC-Jak3-KO mice. This could be due to compromised phagocytic activity of the microglial cells in IEC-Jak3-KO brains, as suggested by the intracellular colocalization in the brain IFC data from these mice, where Jak3f/f (flox control) brains had microglial cells with internalized Aβ (intracellular yellow stain for the TREM2–Aβ complex), indicating the intact phagocytic activity of the control mice. However, the IEC deficiency of Jak3 led to compromised microglial phagocytosis, where the TREM2–Aβ (yellow) complex was decreased.

Lastly, to investigate the mechanistic aspect of how Jak3 regulated the gut–brain axis and dementia pathology, our study focused on two aspects: first, the impact of Jak3 on the gut microbiome, and the second, how Jak3-mediated changes in the gut microbiome impacted the microglial cells in the brain. Interestingly, while it was known that global Jak3 deficiency in general caused obesity-associated metabolic syndrome [20,21], the tissue-specific deficiency of Jak3 in metabolic-syndrome-associated cognitive impairment was not known. To achieve this, we used the previously reported WT and mutants of Jak3 [29] to characterize the role of global Jak3 in the gut–brain communication and used a novel tissue-specific Jak3-KO mouse model to confirm the IEC-specific role of Jak3 in gut–brain communication and obesity/diabetes-associated cognitive impairment characteristic. Our data showed that IEC-Jak3 was essential for the prevention of intestinal-originated systemic obesity-associated glycemic dysregulation and the associated cognitive impairment. IEC prevented these comorbidities by promoting gut symbiosis and intestinal mucosal tolerance by suppressing TLR-mediated inflammatory cytokine production and suppressing neuroinflammation by suppressing the brain TLR-mediated inflammatory cytokine production and suppressing Abeta and pTau accumulation through microglial phagocytosis of them by interacting with and potentially tyrosine phosphorylating microglial TREM2 (Figure 10 model).

Taken together, these results showed, for the first time, the role of intestinal Jak3 signaling in gut–brain communication, where Jak3 mediated gut and brain tolerance through the conserved signaling pathways of TLR4/2 suppression and promoted brain clearance of plaques and tangles through interactions with microglial TREM2. These studies also determined the previously unknown functions of Jak3 in obesity/diabetes-associated cognitive impairment. Thus, the present study demonstrated a novel molecular basis for Jak3 in the gut–brain communication and its physiological and pathophysiological implications in CLGI, obesity-associated glycemic dysregulation, gut dysbiosis and the associated cognitive impairment, which, in future, would have a wider impact on our understanding of the gut–microbiome–brain interactions and dementia.

## 4. Experimental Procedures

### 4.1. Materials

#### 4.1.1. Mice

Six-to-eight-week-old C57BL/6 mice (WT) or C57BL/6-background jak3−/− mice (KO): S form with intact kinase domain were obtained from the Jackson Laboratory (USA). jak3−/− mice were back-crossed to C57BL/6 jak3+/+ to generate jak3−/+ mice, which were then in-bred to generate jak3−/− and jak3+/+ littermates, which were either co-housed or housed separately, according to the sex and genotype. Jak3 KO, intestinal epithelial specific and immune specific Jak3 KO 20 transgenic, have been described. Jak3 KOIEC mice were generated using Jak3+/+ embryonic stem cells to create C57BL/6J-Jak3 f/f mice, which were crossed with villin-Cre mice. Littermate Oclnf/f lacking Cre were used as wild-type controls in the experiments with Jak3 KOIEC mice. For HFD studies, male C57/BL6 mice, weighing 22 ± 2 g, were fed a high-fat diet (65% cal from fat) or a normal diet (6.5% cal from fat) for 8 weeks. The animals were housed in a temperature- and light-controlled room. The mice had food and water ad libitum. Mouse chow was purchased from Research Diets (New Brunswick, NJ, USA). This study adhered to the institutional guidelines of Texas A&M University Institutional Animal Care and Use Committee.

#### 4.1.2. Antibodies

The following antibodies were used in this study: pY20 (MP Biomedicals, Irvine, CA, USA); IgG control (Invitrogen, Thermo Fisher Scientific, Rockford, IL, USA); Pierce^®^ BCA Protein Assay Kit (Thermo Scientific); Jak3, IgG, β-actin, TLR4, villin (Santa Cruz Biotechnology Inc, Dallas, TX, USA), TLR2, TREM 2, Iba1 (Novous Biologicals, Centennial, CO, USA), NF-κB, *p* NF-κB, pAkt, Akt, β–amyloid (Cell signaling technologies, Danvers, MA, USA), FITC(Sigma Spring, TX, USA), Cy3: (Sigma), and P-Tau: (abcam, Waltham, MA, USA)

### 4.2. Methods

#### 4.2.1. Behavioral Studies

Rotarod test. A rotarod test was performed at the age of six months with accelerating rotarod apparatus (IITC Life Science, CA, USA). After a pre-trained period of two days at a constant speed (5 rpm on the first day over 1 min four times, and 8 rpm over 1 min four times on the second day), on the third day, the rotarod accelerated from 5 rpm, the top speed was set at 30 rpm, and the ramp speed was set at 30 s, and mice were tested five times. The latency to fall was measured during the accelerating trials.

Tail-flick test. Animals were placed on the tail-flick apparatus with their tails smoothed into a groove, which contained a photocell. A light source was activated, and the light remained focused on the tail until the mouse moved its tail (a spinal reflex), thus switching the light off. The intensity of the light was adjusted to obtain a baseline tail-flick latency of 2–4 s, and a cut-off time of 9 s was chosen to prevent tissue damage [101].

Hot plate. This was tested by placing male mice (40–50 g) on a hot plate at a constant temperature (52.5 °C) or RT plate twice a day for 2 consecutive days. The mice remained on the room-temperature plate for 50 s at each trial. A control group was moved into the testing room twice a day for 2 days, handled for 50 s and then returned to their home cage. On day 3 (trial 5), the baseline hot-plate latency was recorded for mice from all groups. The temperature of the plate was monitored at all times. To confine the animals to a certain observation area, a colorless acrylic cylinder of 20 cm in diameter was placed on the hot plate. After each measurement, the plate was wiped with a damp cloth to remove traces of urine and feces.

#### 4.2.2. Eight-Arm Radial Maze

The eight-arm radial maze protocol was performed as previously described in the four-arm baited version (Jarrard, 1983, Satoh et al., 2007). Briefly, the apparatus consisted of eight identical arms extending radially from an octagonal platform. It was elevated 80 cm above the floor and surrounded by external cues. The overhead light was on, and 10 stimuli were attached to the curtain around the maze. Five of the stimuli were shapes cut from white cardboard. The shapes were a star, a rectangle, a circle and a triangle. The other five stimuli were a yellow oval, a small red plastic butterfly, a small yellow plastic ball and a black-and-white poster. A cup containing food was placed at the end of each arm. The test was performed in three phases: (1) habituation, which consisted of one exploratory trial (10 min) to prepare the animals for the maze; (2) acquisition, which consisted of two consecutive trials (5 min each) performed once a day over 8 consecutive days; and (3) retention, consisting of unique 5 min trials that were carried out 48, 72, 96, 120 and 144 h after completion of the previous session. The first retention session was carried out 48 h after the last acquisition session; 72 h after the first retention trial, the second retention session was carried out; 96 h after the second retention trial, the third retention session was performed, etc. In the habituation trial, reward pellets (chocolate-flavored food pellets) were placed both at the end and at the entrance of all eight arms. During acquisition sessions, only four arms were baited, and these arms remained baited during all acquisition and retention sessions. For the trials, mice were handled for 30 s, placed on the central platform and allowed to move freely until all pellets were consumed. A 5 min cut-off time period was arbitrarily established. An arm entry was counted when all four paws of the animal crossed the mouth of the arm. A perfect entry was defined as a first entry/food consumed into a baited arm. Entry in a non-baited arm was considered as a reference memory error, and a re-entry in a previously visited baited arm was considered as a working memory error. Total errors were the sum of both reference and working memory errors. Reference errors were not represented in the graphs, instead being included in the total error graphs. Differences among the groups in latency to reach the four baited arms, working, reference and total memory errors were evaluated in each trial.

#### 4.2.3. Fasting Blood Glucose Test, Glucose Tolerance Test and Insulin Tolerance Test

After mice were deprived of food for 4h (before the insulin tolerance test) or overnight (12 h; for fasting glucose), basal blood glucose levels were measured via tail bleeding using a glucometer (free style). For the glucose tolerance test, the mice were intraperitoneally injected with a bolus of glucose (2 mg of glucose per g of body weight). For the insulin tolerance tests, the mice were subjected to a bolus injection of insulin (0.75 m units of insulin/g of body weight). Blood glucose levels were subsequently measured at the times indicated, as reported by the groups.

#### 4.2.4. Tissue Lysate Preparation

Lysates of the frozen colon and brain tissue samples were prepared as reported before [20]. Mice were anesthetized, euthanized, dissected, and colon and brain segments were removed as previously described (R). The tissue was freshly frozen in dry ice and stored at −80. Tissue digestion was performed by sequentially incubating the tissue in the tissue lysis buffer, 1X RIPA Buffer: 20 mM Tris-HCl (pH 7.5) 150 mM NaCl, 1 mM Na2EDTA 1 mM EGTA, 1% NP-40, 1% sodium deoxycholate, 2.5 mM sodium pyrophosphate, 1 mM b-glycerophosphate, 1 mM Na3VO4,1 µg/mL leupeptin. The tissue was mechanically dissociated by gentle trituration, followed by centrifugation. The supernatant was decanted, aliquoted and stored at −80 for further analysis.

#### 4.2.5. Co-Immunoprecipitation (Co-IP), SDS-PAGE and Immunoblotting

Total proteins from the brain and colon cells from the control and genetically modified mice were extracted and incubated overnight with rabbit-IgG or IP-indicated antibody (1 μg) at 4 °C, while an appropriate number of extracted proteins were acted as the input control. Next, 30 μL of Protein A Sepharose was added and incubated for 1 h at 4 °C to form an immune complex. Following centrifugation for 4 min at 3000 rpm at 4 °C, the Protein A Sepharose beads were washed 4 times with 1 mL of washing buffer and boiled in the appropriate protein loading buffer for 5 min. Followed by centrifugation at 3000 rpm, the supernatants were collected into a new tube for Western blot analysis. The antibodies used for IP detection and Western blot analysis are indicated in the Materials section. Blotting was performed on nitrocellulose membranes (GE Healthcare) using standard techniques. The membranes were blocked for 1 h in TBS-0.1% Tween 20 with 5% non-fat dry milk or 1 % polyvinylpyrrolidone-40 and 0.05 % Tween-20 and incubated with primary antibodies overnight. After three washing steps, the membranes were incubated with respective peroxidase-conjugated secondary antibodies, and the signals were recorded by chemiluminescence according to the manufacturer’s protocol (Thermo Scientific, Rockford, IL, USA).

#### 4.2.6. Isolation and Sequencing of Bacteria from Mouse Stool and Taxonomic Identification of Sequenced Isolates

Stool samples from mice were collected by standard protocol. The sample was sent to the commercial sequencing facility. Each repeated run was therefore treated as a technical replicate to determine (i) the measurement error for the estimation of intragenomic 16S gene SNP frequencies attributable to the sequencing platform and (ii) the relationship between the measurement error and sequencing depth. Sequence data for each isolate were quality filtered and adapters removed as described above by the facility.

#### 4.2.7. Isolation of Cells for Flow Cytometry

##### Isolation of Cells from Colon and Brain Tissues

Brain cells isolation. The mice were euthanized by CO2 and transcardially perfused with ice-cold PBS prior to brain extraction. Whole brains, excluding brain stem and olfactory bulbs, were dissected. Single-cell suspensions were prepared from brain tissues by mechanical dissociation using mesh of decreasing sizes, from 250 to 70 μm, and enriched for microglia by density gradient separation [102]. Briefly, the cell pellet was resuspended in 70% (*v*/*v*) isotonic Percoll (1x PBS + 90% (*v*/*v*) Percoll), overlaid with 37% (*v*/*v*) isotonic Percoll and centrifuged with slow acceleration and no brake at 2000 *g* for 20 min at 4 °C. The microglia-enriched cell population isolated from the 37–70% interphase was diluted 1:5 in ice-cold PBS and recovered by cold centrifugation at maximum speed for 1 min in microcentrifuge tubes. The cell pellet was then stained with antibodies to microglial cell surface markers (CD11b-BV650, 1:200 Biolegend, #141723; CD11a, 1:20, BD Biosciences, #558191, TREM2-APC, 1:10, R&D Systems, #FAB17291N;) for analysis using the BD-Accuri C6.

Colonic cells isolation. Colons were dissected out from mice, and fat and blood vessels were removed from the colon. Colons were cut open longitudinally and washed with PBS to remove feces and debris, then incubated in PBS containing 5 mM EDTA, 0.145 mg/mL dithiothreitol (Sigma-Aldrich), 3% FBS and 1% penicillin/streptomycin (P/S) for 15 min at 37 °C for 2 cycles. After being vortexed for 15 s, the dissociated cells were collected as colonic epithelial cells. For the isolation of lamina propria immune cells, the remaining colonic tissues were washed twice in PBS, cut into 1 mm in length and digested in RPMI 1640 containing 0.5 mg/mL collagenase D (Roche), 0.01 mg/mL DNase I (Roche) and 0.5 mg/mL dispase (Stem Cell Technologies) for 30 min at 37 °C on a shaking platform. The digested tissues were passed through 70 μm strainers after being vigorously vortexed for 15 s. Then, the colonic immune cells were collected and resuspended in a staining buffer (PBS with 1% fetal bovine serum and 1% penicillin-streptomycin solution (Corning)) for flow cytometry analysis.

Flow cytometry. Single-cell suspensions were stained with a combination of fluorescently conjugated monoclonal antibodies. CD16/32 antibody (93; BioLegend) was used to block the non-specific binding to Fc receptors before surface staining. Cells were stained with fixable yellow dead cell stain kit (Invitrogen) for the detection of live/dead cells before staining of the cell surface. All antibodies were purchased from BioLegend unless otherwise specified. For surface marker staining, we used the antibodies to the indicated mouse proteins.

In simple terms, cell suspensions were washed and re-suspended in 1:100 dilutions of primary antibodies in a solution containing PBS with 1% BSA and 0.01% sodium azide for all the experimental conditions. A secondary fluorophore conjugate of FITC or CY3 were used subsequent to incubation with the primary antibody. The experimental samples were incubated at room temperature for 30 min and washed in PBS with 1% BSA and 0.01% sodium azide after incubation with primary and secondary antibodies. Flow cytometric data were acquired in FL 1 (FITC, 533/30) and FL 2 (Cy3, 585/40) channels using BD Accuri C6 flow cytometer (BD biosciences, CA, USA) and analyzed using BD Accuri C6 software with appropriate unstained and gating controls. The percentages of cells determined to be positive on the FL1 or FL 2 channels were plotted in the form of a bar graph with Origin 8.6 (Originlab, Washington, DC, USA) software and statistically compared for all the treatment groups.

##### Immunohistochemistry

For IFM, the OCT mold embedded tissue sections were air dried for 20 min at RT, fixed using 4% paraformaldehyde and blocked using 5% BSA in PBS for 30 min. Sections were then incubated with primary antibodies for Jak3, TLR4, TLR2, villin, F4/80, pNF-κB, NF-κB, Pakt, AKT and Jak3 pTau, β- amyloid followed by incubation with cy3 or Alexa-fluor 488 conjugated secondary antibodies. The sections were than rinsed twice with PBS and mounted using Vectashield (Vector Lab, Burlingame, CA, USA). For all negative controls, primary antibodies were replaced with a control non-immune IgG at the same concentrations. IFM for Caco2 cells transduced with RFP-tagged Jak3 shRNA were performed as reported previously. The immunostained slides were visualized using C1-plus Nikon laser scanning confocal microscope, and the images were processed using NIS element software (Nikon^R^). All experiments were conducted at least in triplicate, and representative images were shown.

##### Statistics

Data are expressed as mean +/− standard error of mean and were analyzed using the Mann–Whitney test, *t*-test and one-way analysis of variance (ANOVA). Significance was deemed when the *p* value was less than 0.05. Statistical analysis was performed with GraphPad Prism 5 (GraphPad Software Inc, La Jolla, CA, USA). All data are presented as S.E. and analyzed using Microcal Origin^®^ software version 9.2. Differences in the parametric data were evaluated by the Student’s t-test. Significance in all tests was set at a 95% or greater confidence level. Statistically significant data and the corresponding *p*-value are annotated in the figure legends.

## Figures and Tables

**Figure 1 nutrients-14-03715-f001:**
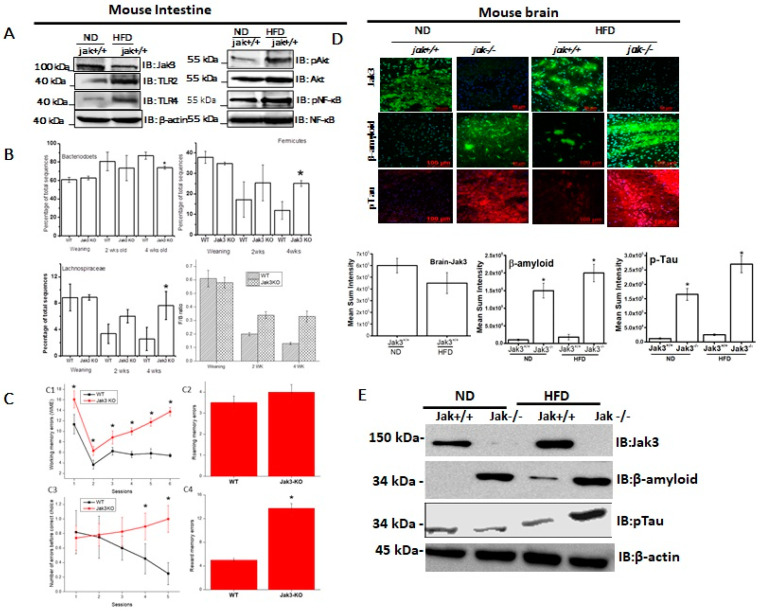
High-fat diet (HFD) reduces intestinal expression of Jak3, and Jak3 deficiency in mice leads to colonic dysbiosis, cognitive impairment and accumulation of β-Amyloid and pTau in obese mouse brain. (**A**) Colons from normal diet (ND) and HFD-fed mice were excised out, and Western blot analysis was performed using the tissue lysates for the indicated proteins using β-actin as controls. Representative blots (*n* = 3) are shown. (**B**) Jak3 deficiency promotes gut dysbiosis. Gut microbiome composition was determined using a commercial facility through 16sRNA pyrosequencing of fecal DNA samples from co-housed WT and Jak3-KO littermates (*n* = 10 each group). The time courses of changes in fecal microbiota from WT and Jak3-KO mice are shown through the shift in the relative abundance of specific taxa (Upper panels; UP), specific family (Lower left panel; LLP), and the time courses of the relative shift in Firmicutes to Bacteroidetes ratio (F/B) (Lower right panel; LRP) are shown. Statistical analysis was performed using repeated measures paired group analysis of variance. Error bars represent +/−SEM. * indicate statistically significant differences compared to WT (ULP; *p* = 0.004, URP; *p* = 0.002, LLP; *p* = 0.001). (**C**) Jak3 deficiency promotes cognitive impairment. Automated elevated radial arm maze equipped with MazesoftTM Software was used to calculate the four parameters of cognitive assessments, viz., working memory errors (WME) (Upper left), roaming memory errors (RME) (Upper right), number of errors before a correct choice (EBCC) (Lower left) and reward memory errors (RWME) (Lower Right) over five sessions. A repeated-measure ANOVA on performances of age- and sex-matched wild-type (WT) littermate controls of Jak3-KO mice (*n* = 20 each group) are shown for each session (Upper and lower left panels) or an average of all the sessions (Upper and lower right panels). All the data are representative of three independent experiments. Values are mean ± S.D. * denotes *p* < 0.05 compared with WT mice. (Upper left; *p* = 0.002, Lower left; *p* = 0.004, Lower right; *p* = 0.001). (**D**) Jak3 deficiency promotes accumulation of β-Amyloid and pTau in mouse brain. Brain tissue sections from WT and corresponding Jak3-KO littermate mice fed with either ND or HFD were immunostained using β-amyloid or pTau or Jak3 primary antibodies followed by FITC- or Cy-3-conjugated secondary antibodies. Mounting media containing PI were used to visualize the nucleus. Representative images (*n* = 10) are shown (Lower panels). Quantifications of the intensities in the “upper panels” were performed using NikonR C1-plus imaging software, and the results were normalized against PI for Jak3 (Lower left), β-amyloid (Lower middle) and pTau (Lower right). Values are mean ± S.D. Asterisk denotes *p* < 0.05 compared with WT mice group. (Jak3 *p* = 0.003, β-amyloid *p* = 0.005, pTau *p* = 0.002). (**E**) Brain tissue lysates from the samples in “D” were used to perform IB in the presence of the indicated antibodies with β-actin as control. Representative blots (*n* = 3) are shown.

**Figure 2 nutrients-14-03715-f002:**
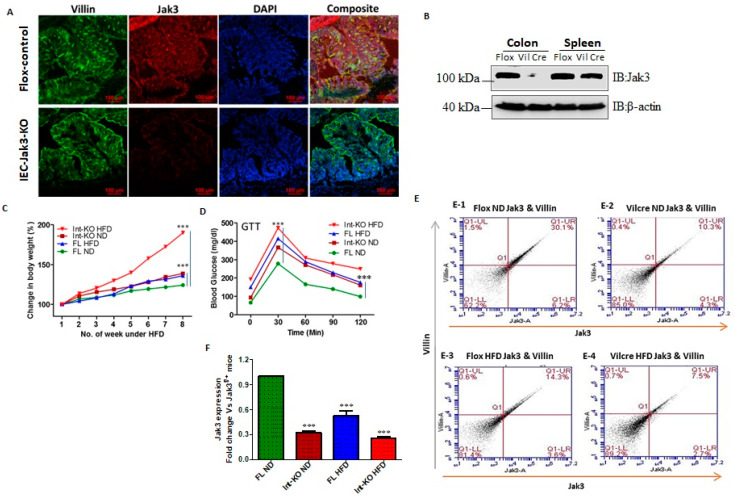
Intestinal epithelial cell (IEC) deficiency of Jak3 predisposes to exaggerated symptoms of HFD-induced obesity and glycemic dysregulation. (**A**), Genetic manipulation shows IEC deficiency of Jak3. Colonic tissue sections from floxed Jak3 (jak3f/f) or IEC-Jak3-KO (vil1-Cre- jak3f/f) were co-immunostained with IEC marker villin (green) and Jak3 (red) antibodies using DAPI as control for the nucleus. Representative images (*n* = 10) are shown from each group (*n* = 6). (**B**), Western blot analyses were performed using tissue lysates from floxed Jak3 and IEC-Jak3-KO mice for Jak3 expression (upper panel) using β-actin as control (lower panel). Representative blots (*n* = 3) are shown. (**C**), Fluorescence-activated cell sorting (FACS) analysis is presented as four quadrant dot plots to determine the impact of HFD on IEC-Jak3 expression by determining the double positive cells (Quadrant:UR) for IEC marker villin and Jak3 under ND or HFD. Jak3 positive cells are shown on the x axis and those of villin positive cells on the y axis. (**D**), Bar chart results following FACS experiments in “C” were repeated (*n* = 5), and mean ± SD values are shown as bar graph for the comparative average cell counts for the indicated groups of mice. *** denotes *p* = 0.004 (Int-KO ND), *p* = 0.008 (FL HFD), *p* = 0.001(Int-KO HFD). (**E**), IEC deficiency of Jak3 predisposes to increased body weight. Vil1-Cre/ jak3f/f (Int-KO) or their littermate controls jak3f/f (FL) mice were cohoused and subjected to either ND or HFD. Body weights of co-housed Int-KO and their FL littermate males were plotted at the indicated intervals as a percentage of their body weight at the time of weaning (*n* = 5/group). *** denotes *p* = 0.005 (Int-KO HFD), *p* = 0.05 (Int-KO ND). (**F**), IEC deficiency of Jak3 impairs the ability to restore blood glucose. Blood glucose concentrations in fasted mice were measured after intraperitoneal injection with 2 g/kg body weight of d-glucose followed by measuring the level of blood glucose at thew, indicating intervals post-injection (*n* = 5/group). *** denotes *p* = 0.003 (Int-KO HFD), *p* = 0.05 (Int-KO ND).

**Figure 3 nutrients-14-03715-f003:**
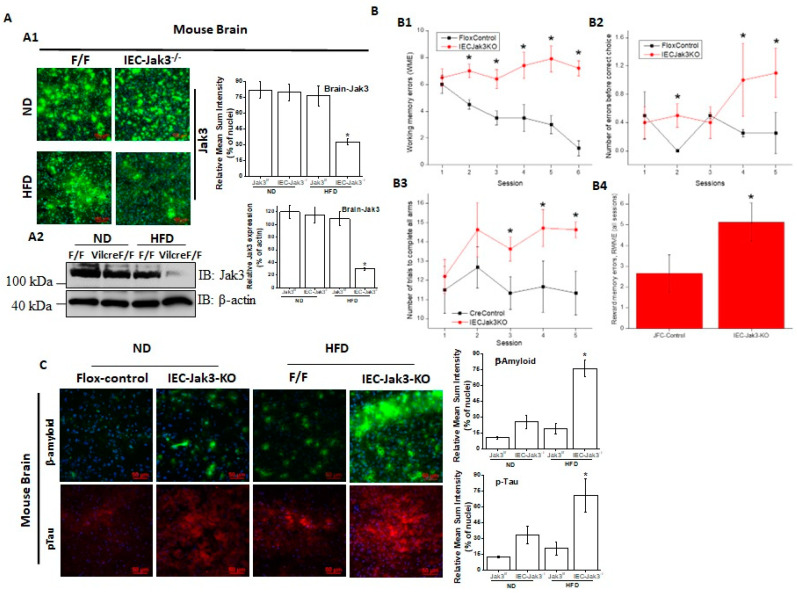
IEC deficiency in Jak3 is responsible for cognitive impairment and increased cerebral cortex accumulation of Aβ and pTau during HFD-induced obesity: (**A**) IEC-Jak3 deficiency promotes HFD-mediated impacts on Jak3 expression in the brain in mice. Brain tissue sections (**A1**) or tissue-lysates (**A2**) from FL-control and corresponding IEC-Jak3-KO littermate mice fed with either ND or HFD were immunostained or Western blotted using Jak3 primary antibodies followed by FITC- or HRP-conjugated secondary antibodies, respectively. Mounting media containing PI were used to visualize the nucleus in A1. Representative images in A1 or blots in A2 (*n* = 10 and 3, respectively) are shown (Right panels). Quantifications of the florescent intensities in A1 or densitometric analyses in A2 in the corresponding “left panels” were performed using NikonR C1-plus imaging and BioRad software, respectively, and the results were normalized against PI in B1 or β-actin in B2 for Jak3 expression. (**B**) IEC-Jak3 deficiency promotes cognitive impairment during obesity. The four parameters of cognitive assessments, viz., WME (**B1**), EBCC (**B2**), RME (**B3**) and RWME (**B4**), were measured, as in Figure 1C. A repeated-measure ANOVA on performances of age- and sex-matched Jak3-flox (Flox-control) and littermate controls of IEC-Jak3-KO mice (*n* = 10 each group) is shown for each session (**B1–3**) or an average of all the sessions (**B4**). Data are representative of three independent experiments. Values are mean ± S.D. * denotes statistically significant compared with flox mice. (A1; *p* > 0.01, A2; *p* > 0.05, A3; *p* > 0.01, A4). (**C**) IEC-Jak3 deficiency promotes accumulation of β-Amyloid and pTau in mouse brain. Brain tissue sections from FL-control and corresponding IEC-Jak3-KO littermate mice fed with either ND or HFD were immunostained using β-amyloid or pTau or Jak3 primary antibodies followed by FITC- or Cy-3-conjugated secondary antibodies. Mounting media containing PI were used to visualize the nucleus. Representative images (*n* = 10) are shown (Right panels). Quantifications of the intensities in the “left panels” were performed using NikonR C1-plus imaging software, and the results were normalized against PI for β-amyloid (Upper right) and pTau (Lower right). Values are mean ± S.D. Asterisk denotes statistically significant *p* > 0.05 compared with flox mice group. (β-amyloid *p* = 0.005, pTau *p* = 0.003).

**Figure 4 nutrients-14-03715-f004:**
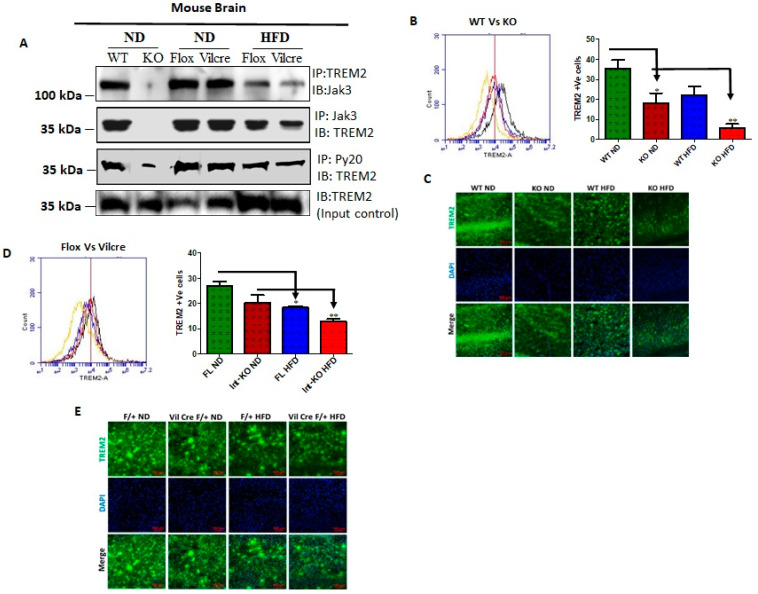
The triggering receptors on microglial cells 2 (TREM2) interact with Jak3 in the brain, and the intestinal deficiency of Jak3 leads to reduced interactions and expression of TREM-2 in the brain with Jak3 during HFD-induced obesity: (**A**) Jak3 interactions with TREM-2 were determined using brain tissue lysates from ND- and HFD-fed mice. Lysates from WT (littermate control), Jak3-KO, IEC-Jak3-KO and flox-Jak3 (littermate control) were subjected to IP followed by IB using the indicated antibodies. Blots shown represent *n* = 3 experiments. (**B**) The impact of global and intestinal epithelial tissue-specific deficiencies of Jak3 on the brain expression of β-Amyloid receptor TREM2 on microglial cells was determined in ND- and HFD-fed mouse brain. Representative flow cytometric histogram graphs of individual mouse brain cells showing the microglial levels of expression of TREM-2 receptor in the four groups (*n* = 5/group; WT-ND (littermate control), Jak3-KO-ND, WT-HFD (littermate control), Jak3-KO-HFD) are shown in the left panel, and the corresponding histogram bar graphs indicating mean ± SD values are shown for the comparative average cell counts for the indicated groups of mice. * Indicate statistically significant difference from the corresponding controls (KO-ND *p* = 0.05, KO-HFD *p* = 0.03). ** Comparison between Int-KO ND and Int-KO HFD group. (**C**) Brain tissue sections from WT-control littermate and Jak3-KO mice fed with either ND or HFD were immunostained using TREM-2 primary antibodies followed by FITC secondary antibodies, and mounting media containing PI were used to visualize the nucleus. Representative images (*n* = 10) are shown. (**D**) Data from similar experiments as in “B” but for the four different groups of mice (*n* = 5/group; flox-Jak3-ND (littermate control), IEC-Jak3-KO-ND, flox-Jak3-HFD (littermate control), IEC-Jak3-KO-HFD) are shown. * Indicate statistically significant difference from the corresponding controls (IEC-Jak3-KO-ND *p* = 0.05, KO-HFD *p* = 0.05). (**E**) Brain tissue sections from flox-Jak3-control littermate and IEC-Jak3-KO mice fed with either ND or HFD were immunostained using TREM-2 primary antibodies followed by FITC secondary antibodies, and mounting media containing PI were used to visualize the nucleus. Representative images (*n* = 10) are shown.

**Figure 5 nutrients-14-03715-f005:**
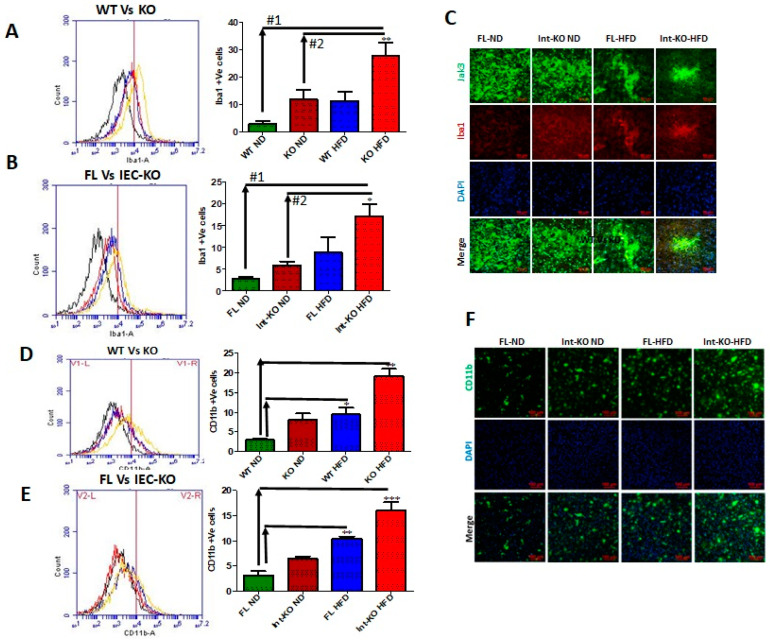
Intestinal deficiency of Jak3 leads to increased microglial activation in the brain during HFD-induced obesity. The impact of global (**A**,**D**) and intestinal epithelial tissue-specific (**B**,**E**) deficiencies of Jak3 on brain expression of microglial marker TREM-2 was determined in ND- and HFD-fed mouse brain. Representative flow cytometric histogram graphs of individual mouse brain cells showing the microglial levels of expression of Iba1 in the four indicated groups of global deficiency or IEC deficiency, respectively (*n* = 5/group), are shown in the left panel, and the corresponding histogram bar graphs indicating mean ± SD values in the right panel are shown for the comparative average cell counts for the indicated groups of mice. *, **, *** Indicate statistically significant difference from the corresponding controls ((**A**): KO-HFD#1 *p* = 0.05, KO-HFD#2 *p* = 0.03; (**B**): IEC-KO-HFD#1 *p* = 0.05, IEC-KO-HFD#2 *p* = 0.04; (**D**): WT-HFD *p*= 0.05, KO-HFD *p* = 0.03; (**E**): FL-HFD *p* = 0.07, IEC-KO-HFD *p* = 0.04). (**C**) Brain tissue sections from flox-Jak3-control littermate and IEC-Jak3-KO mice fed with either ND or HFD were immunostained using microglial marker Iba1 primary antibodies followed by Cy3 secondary antibodies, and mounting media containing PI were used to visualize the nucleus. Representative images (*n* = 10) are shown. (**E**) Brain tissue sections from flox-Jak3-control littermate and IEC-Jak3-KO mice fed with either ND or HFD were immunostained using microglial activation marker CD11b primary antibodies followed by FITC secondary antibodies, and mounting media containing PI were used to visualize the nucleus. (**F**) Similar experiments were performed as in “C” except the brain tissue sections were immunostained using microglial activation marker CD11b primary antibodies followed by FITC-conjugated secondary antibodies. (C&F) Representative images (*n* = 10) are shown.

**Figure 6 nutrients-14-03715-f006:**
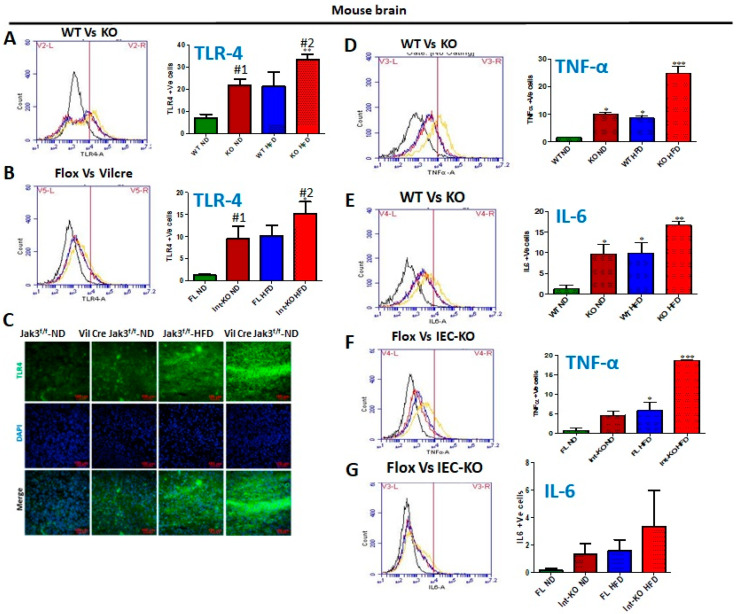
HFD-led suppression of Jak3 promotes brain inflammation through microglial activation and increased TLR4 signaling. Global (**A**,**D**,**E**) or intestinal epithelial (**B**,**F**,**G**) deficiency of Jak3 leads to increased TLR4 expression and inflammation in the brain. Representative flow cytometric histogram graphs of individual mouse brain cells showing the levels of expression of TLR-4 (**A**,**B**) and inflammatory cytokines TNF-α (**D**,F) and IL-6 (**E**,**G**) in the four indicated groups of global Jak3 deficiency or IEC Jak3 deficiency, respectively (*n* = 5/group), are shown in the left panel, and the corresponding histogram bar graphs indicating mean ± SD values in the right or lower panels are shown for the comparative average cell counts for the indicated groups of mice. *, **, *** Indicate statistically significant difference from the corresponding controls (A: KO-ND#1 *p* = 0.04, KO-HFD#2 *p* = 0.07; B: IEC-KO-ND#1 *p* = 0.05, IEC-KO-HFD#2 *p* = 0.05; D: KO-ND *p* = 0.03, WT-HFD *p* = 0.01, KO-HFD *p* = 0.02; E: KO-ND *p* = 0.03, WT-HFD *p* = 0.01, KO-HFD *p* = 0.02; F: FL-HFD *p* = 0.06, IEC-KO-HFD *p* = 0.02)). (**C**) Brain tissue sections from flox-Jak3-control littermate and IEC-Jak3-KO mice fed with either ND or HFD were immunostained using TLR4 primary antibodies followed by FITC secondary antibodies, and mounting media containing PI were used to visualize the nucleus. Representative images (*n* = 10) are shown.

**Figure 7 nutrients-14-03715-f007:**
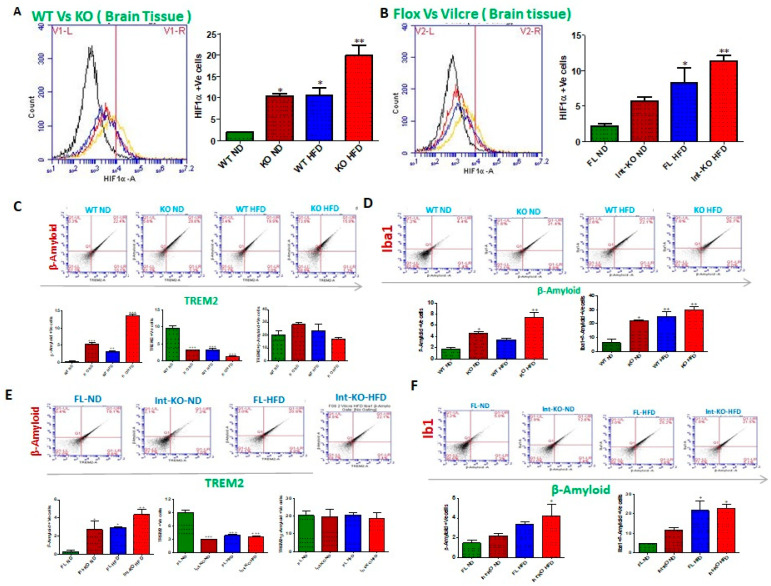
Intestinal epithelial deficiency of Jak3 causes Brain-hypoxia led increase in microglial accumulation of Abeta in the brain. (**A**,**B**), Representative flow cytometric histogram graphs of individual mouse brain cells showing the levels of expression of hypoxia inducible factor 1-α (HIF1-α) in the four indicated groups of global Jak3 deficiency (**A**) and IEC Jak3 deficiency (**B**), respectively (*n* = 5/group), are shown in the left panel, and the corresponding histogram bar graphs indicating mean ± SD values in the right panels are shown for the comparative average cell counts for the indicated groups of mice. *, **, *** Indicate statistically significant difference from the corresponding controls (A: KO-ND#1 *p* = 0.06, WT-HFD#2 *p* = 0.05 KO-HFD#3 *p* = 0.04; B: Int-KO-ND#1 *p* = 0.06, FL-HFD#2 *p* = 0.05, Int-KO-HFD#3 *p* = 0.05). (**C**,**D**), FACS analysis is presented as four quadrant dot plots to determine the impact of global Jak3 deficiency on microglial accumulation of Abeta in the individual mouse brain tissues from “A” by determining the double positive cells (Quadrant:UR) for microglial Abeta receptor TREM-2 on the X axis and Abeta on the Y axis (**C**) and microglial marker Iba1 on the X axis and Abeta on the Y axis (**D**) under ND or HFD. The lower panels in “C” and “D” show the corresponding bar chart results following repeating (*n* = 5) the FACS experiments, and mean ± SD values are shown as bar graph for the comparative average cell counts for the indicated groups of mice. *, **, *** denote *p* < 0.04 in all groups. (**E**,**F**), similar strategies as in C-D were used to perform FACS analysis to determine the impact of intestinal epithelial Jak3 deficiency on microglial accumulation of Abeta in the individual mouse brain tissues from “B” in the indicated four groups under ND or HFD. *, **, *** denote statistically significant values in the indicated group assumed at *p* < 0.05.

**Figure 8 nutrients-14-03715-f008:**
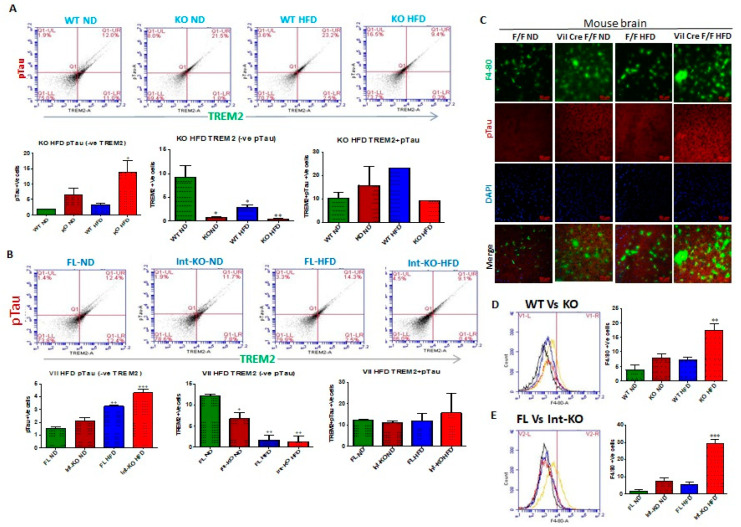
Intestinal epithelial cell deficiency of Jak3 causes increased microglial activation associated microglia accumulation of pTau in the brain. (**A**), FACS analysis is presented as four quadrant dot plots to determine the impact of global Jak3 deficiency on microglial accumulation of pTau in the individual mouse brain tissues by determining the double positive cells (Quadrant:UR) for microglial pTau receptor TREM-2 on the X axis and pTau on the Y axis in mice fed with either ND or HFD. The lower panels show the corresponding bar chart results following repeating (*n* = 5) the FACS experiments and mean ± SD values are shown as bar graph for the comparative average cell counts for the indicated groups of mice. *, ** denotes *p* < 0.04 in all groups. (**B**), similar strategy as in “A” was used, except using intestinal epithelial Jak3-deficient mouse brains to determine microglial accumulation of Abeta in the individual mouse brain tissues from the indicated four groups under ND or HFD. *, **, *** denotes statistically significant values in the indicated group assumed at *p* < 0.05. (**C**), Brain tissue sections from flox-Jak3-control littermate and IEC-Jak3-KO mice fed with either ND or HFD were immunostained using either microglial activation marker F4/80 or pTau primary antibodies followed by FITC and Cy3 secondary antibodies, respectively, and mounting media containing PI were used to visualize the nucleus. Representative images (*n* = 10) are shown. (**D**,**E**), Representative flow cytometric histogram graphs of individual mouse brain cells showing the levels of expression of F4/80 in the four indicated groups of global Jak3 deficiency (**D**) and IEC Jak3 deficiency (**E**), respectively (*n* = 5/group), are shown in the left panels, with the corresponding histogram bar graph indicating mean ± SD values in the right panels for the comparative average cell counts for the indicated groups of mice. **, *** denotes statistically significant values in the indicated group assumed at *p* < 0.05.

**Figure 9 nutrients-14-03715-f009:**
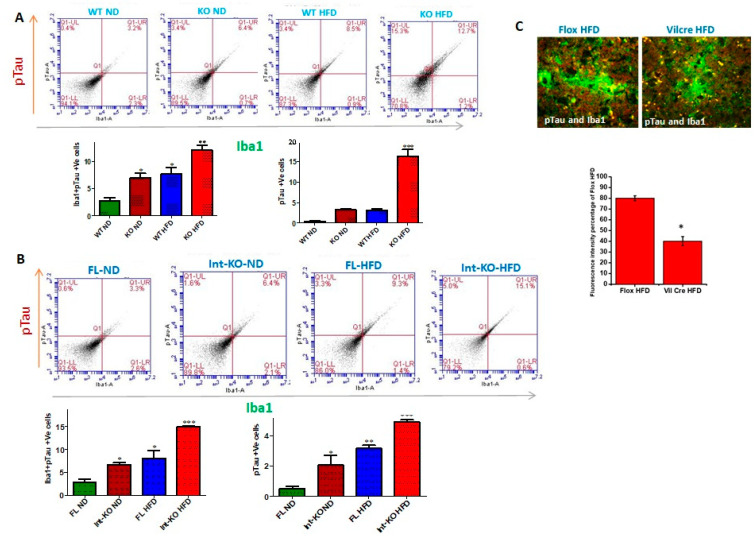
Intestinal epithelial cell deficiency of Jak3 causes suppressed microglial Iba1 associated increased accumulation of pTau in the brain. (**A**,**B**), FACS analysis is presented as four quadrant dot plots to determine the impact of either global Jak3 deficiency (**A**) or IEC Jak3 deficiency (**B**) on microglial accumulation of pTau in the individual mouse brain tissues by determining the double positive cells (Quadrant:UR) for microglial marker Iba1 on the X axis and pTau on the Y axis in mice fed with either ND or HFD. The lower panels show the corresponding bar chart results following repeating (*n* = 5) the FACS experiments, and mean ± SD values are shown as bar graph for the comparative average cell counts for the indicated groups of mice. *, **, *** denotes *p* < 0.04 in all groups. (**C**) Microglial internalization of pTau is compromised in IEC-Jak3-deficient mice. Brain tissue sections from flox-Jak3-control littermate and IEC-Jak3-KO mice fed with HFD were co-immunostained using microglial marker Iba1 and pTau primary antibodies followed by FITC and Cy3 secondary antibodies, respectively, and merged images are shown to visualize the colocalized (yellow) cells. Representative images (*n* = 10) are shown (bottom bar graph). Nikon NIS elementR was used to count the double positive cells, and mean ± SD values are shown as bar graph for the comparative average double positive cells for the indicated groups of mice. * and ** denotes *p* < 0.04 in all groups.

**Figure 10 nutrients-14-03715-f010:**
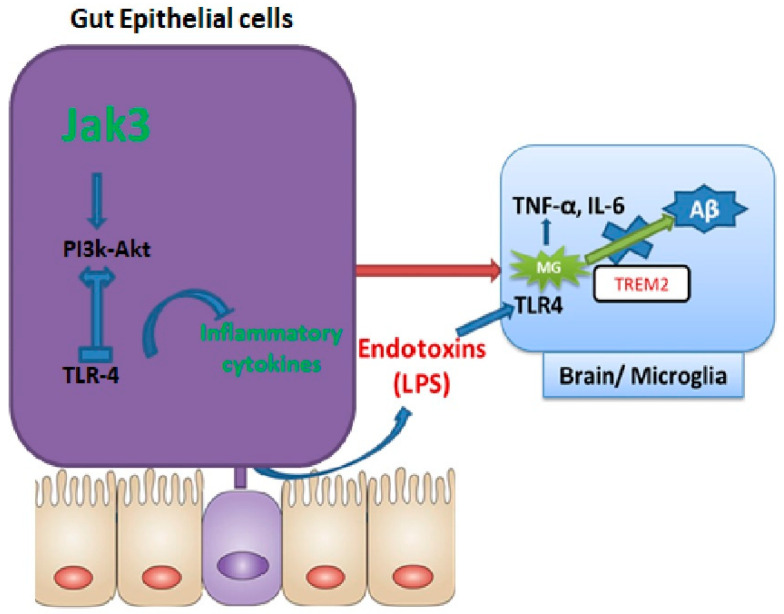
Model for Jak3 signaling in gut–brain axis. Conserved Jak3-mediated signaling in intestinal epithelial cells and microglial cells in the brain are shown.

## Data Availability

The data generated and analyzed during the current study are not publicly available, but are available from the corresponding author on reasonable request.

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
