# Peer review of "Mechanistic Role of Jak3 in Obesity-Associated Cognitive Impairments"

_nutrients, 2022, doi:10.3390/nu14183715_

Round 1

Reviewer 1 Report

In their manuscript entitled "Mechanistic role of Jak3 in obesity associated cognitive impairment" the authors have shown the mechanism by which Jak3 regulates the gut-brain axis resulting in obesity and cognitive impairment. It is an exciting topic and the research is executed logically. 

A few of my comments are as below:

1) Gut microbiome analysis was performed only in the first study. It would be interesting to see the effect of Jak3 ko in the intestine on the gut microbiome.

2)  Lines 869-871 "An impairment in the microbiota-gut-brain (MGB) axis has been suggested in several neurodegenerative diseases with cognitive dysfunction 83, 84 where the circulating gut-microbial products has been implicated for causative role." 

In your study, were these circulating gut-microbial products looked at to show a more direct relation of these metabolites with cognitive function?

3) The word deficiency has been repeatedly misspelled.

Author Response

Rv#1

We would like to thank the reviewers for the review of our manuscript and grateful for their thoughtful feedback. A point-by-point responses to the reviewer’s concerns are provided below;

  • Gut microbiome analysis was performed only in the first study. It would be interesting to see the effect of Jak3 ko in the intestine on the gut microbiome.

Thank you for this excellent suggestion and in fact we are working on the detailed analysis of the changes in gut microbiome under different conditions which include the impact of Jak3-deficiency in the specific cell-types of gut (goblet-cells, enterocytes, enteroendocrine-cells etc.). Since the objective of the current manuscript is to first demonstrate how global deficiency of Jak3 contribute to obesity associated cognitive impairment and then find out which tissue specific Jak3 contribute to such pathology. Since the detailed analysis of gut microbiome during intestinal Jak3 deficiency is a part of another study that we are currently conducting (separate manuscript to be communicated very soon) and is beyond the scope of current manuscript, for that reason, we have not included those in this manuscript.

  • Lines 869-871 "An impairment in the microbiota-gut-brain (MGB) axis has been suggested in several neurodegenerative diseases with cognitive dysfunction 83, 84 where the circulating gut-microbial productshas been implicated for causative role." In your study, were these circulating gut-microbial products looked at to show a more direct relation of these metabolites with cognitive function?

Yes, we did see a 5-fold increase in circulating LPS during global and IEC-Jak3 deficiency and have now mentioned in the discussion section (page 24 Ln 876-877). 

  • The word deficiency has been repeatedly misspelled.

We apologize the reviewer for this oversight and have now revised the manuscript accordingly. 

Reviewer 2 Report

Improve the way of writing and expression

Way of referencing needs to be corrected throughout the article 

A study by Mishra et al (2015) showed ' that Jak3 played a critical role in the pathogenesis of obesity and associated metabolic syndrome under basal conditions and during those induced by high-fat diets where colonic expression of Jak3 was essential for the maintenance of a healthy mucosal barrier, reduced infiltration of macrophages and neutrophils in colonic tissue, and overall mucosal tolerance. Mechanistically, we demonstrated that Jak3 facilitated epithelial tolerance toward LPS-induced increased expression of TLR4 and associated activation of NF-κβ through interactions with and tyrosine phosphorylation of p85, the regulatory subunit of PI3K. Thus, these results showed for the first time the essential role of Jak3 in prevention of obesity and associated MetS where Jak3 facilitated prevention of CLGI through several mechanisms including those through mucosal tolerance toward commensal gut-microbiota by suppressed expression and limited activation of TLRs.' Your reflection to this ?

Author Response

Rev#2

We would like to thank the reviewers for the review of our manuscript and grateful for their thoughtful feedback. A point-by-point responses to the reviewer’s concerns are provided below;

Way of referencing needs to be corrected throughout the article.

We have revised the referencing throughout the article as suggested using the “Numbered” format in EndNote. Since the downloaded manuscript did not allow reformatting the references, for that reason, we have sent a separate word.doc manuscript to Ms. Leslie Duan, the Assistant Editor.

Study by Mishra et al (2015) showed ' that Jak3 played a critical role in the pathogenesis of obesity and associated metabolic syndrome under basal conditions and during those induced by high-fat diets where colonic expression of Jak3 was essential for the maintenance of a healthy mucosal barrier, reduced infiltration of macrophages and neutrophils in colonic tissue, and overall mucosal tolerance. Mechanistically, we demonstrated that Jak3 facilitated epithelial tolerance toward LPS-induced increased expression of TLR4 and associated activation of NF-κβ through interactions with and tyrosine phosphorylation of p85, the regulatory subunit of PI3K. Thus, these results showed for the first time the essential role of Jak3 in prevention of obesity and associated MetS where Jak3 facilitated prevention of CLGI through several mechanisms including those through mucosal tolerance toward commensal gut-microbiota by suppressed expression and limited activation of TLRs.' Your reflection to this?

Current study shows how Jak3-mediated pathways link gut-brain axis by specifying the redundancy in Jak3-mediated tolerogenic impact in intestinal epithelial and microglial cells where HFD promoted Jak3-suppression mediated activation of TLR-signaling and associated inflammation in intestine and in brain. Moreover, in brain microglial cells, Jak3 also facilitated TREM-2-phosphorylation and clearance of Abeta/pTau plaques and a compromise in Jak3-signlaing resulted in obesity associated metabolic syndrome cognitive impairment.